# Lifelong Learning of Video Diffusion Models From a Single Video Stream

## Abstract

This work demonstrates that training autoregressive video diffusion models from a single video stream—resembling the experience of embodied agents—is not only possible, but can also be as effective as standard offline training given the same number of gradient steps. Our work further reveals that this main result can be achieved using experience replay methods that only retain a subset of the preceding video stream. To support training and evaluation in this setting, we introduce four new datasets for streaming lifelong generative video modeling: *Lifelong Bouncing Balls*, *Lifelong 3D Maze*, *Lifelong Drive*, and *Lifelong PLAICraft*, each consisting of one million consecutive frames from environments of increasing complexity.[1] Together, our datasets and investigations lay the groundwork for video generative models and world models that continuously learn from single-sensor video streams rather than from fixed, curated video datasets.

## 1 Introduction

There are a plethora of names – lifelong learning, continual learning, streaming inference – given to a central desideratum of artificial intelligence (AI) systems: the ability to learn from a single continuous autocorrelated stream of data. However named, our community has long sought models and algorithms that learn in a fundamentally human way; from birth to death, learning as we live.

Modern AI systems do not learn in this manner but instead rely on an effective compromise: stochastic gradient descent from data streams made up of independently and identically distributed (i.i.d.) samples. Language models (Touvron et al., 2023; Mukherjee et al., 2023), world models (Hafner et al., 2020; 2021; Brooks et al., 2024; Agarwal et al., 2025), and video models (Harvey et al., 2022; Ho et al., 2022; Bar-Tal et al., 2024) are trained on random batches of short temporally correlated segments, an approach that preserves short-range autocorrelations while approximating i.i.d.-ness through permutations. While effective, this approximately i.i.d. learning paradigm does not offer a satisfying mechanism for updating models when new data arrive. Existing model updating techniques that are often used in practice—training from a checkpoint on the union of old and new data (Ash & Adams, 2020), fine-tuning on just the new data (Xu et al., 2023), or training completely anew from scratch (Ren et al., 2021) – do not operate in the training regime from which humans learn and suffer from problems such as high computation cost and forgetting (Verwimp et al., 2024).

Alternatively, SGD on autocorrelated data streams is considered by some to be a viable candidate for human-like lifelong learning (Lillicrap et al., 2020). While there is a raft of work indicating that gradient-based learning on autocorrelated data is possible (Duchi et al., 2012; Johansson et al., 2010; Ram et al., 2009), folk wisdom maintains that this learning setup is hard, impractical, and prone to failure, especially for deep networks. Evidence of these beliefs can be found throughout the literature. The optimization community has developed numerous mechanisms for alleviating the effects of data stream temporal dependencies (Kowshik et al., 2021; Godichon-Baggioni et al., 2023; Chang & Shahrampour, 2022), suggesting the existence of problems with learning from autocorrelated data streams. The Bayesian learning community has proposed numerous continual learning approaches through posterior updating (Bartlett & Wood, 2011; Broderick et al., 2013; Naesseth et al., 2019; Beronov et al., 2021), though these approaches lack practical scaled results.

---

[1]Code and datasets will be released on acceptance. Video samples are available in the supplementary materials.

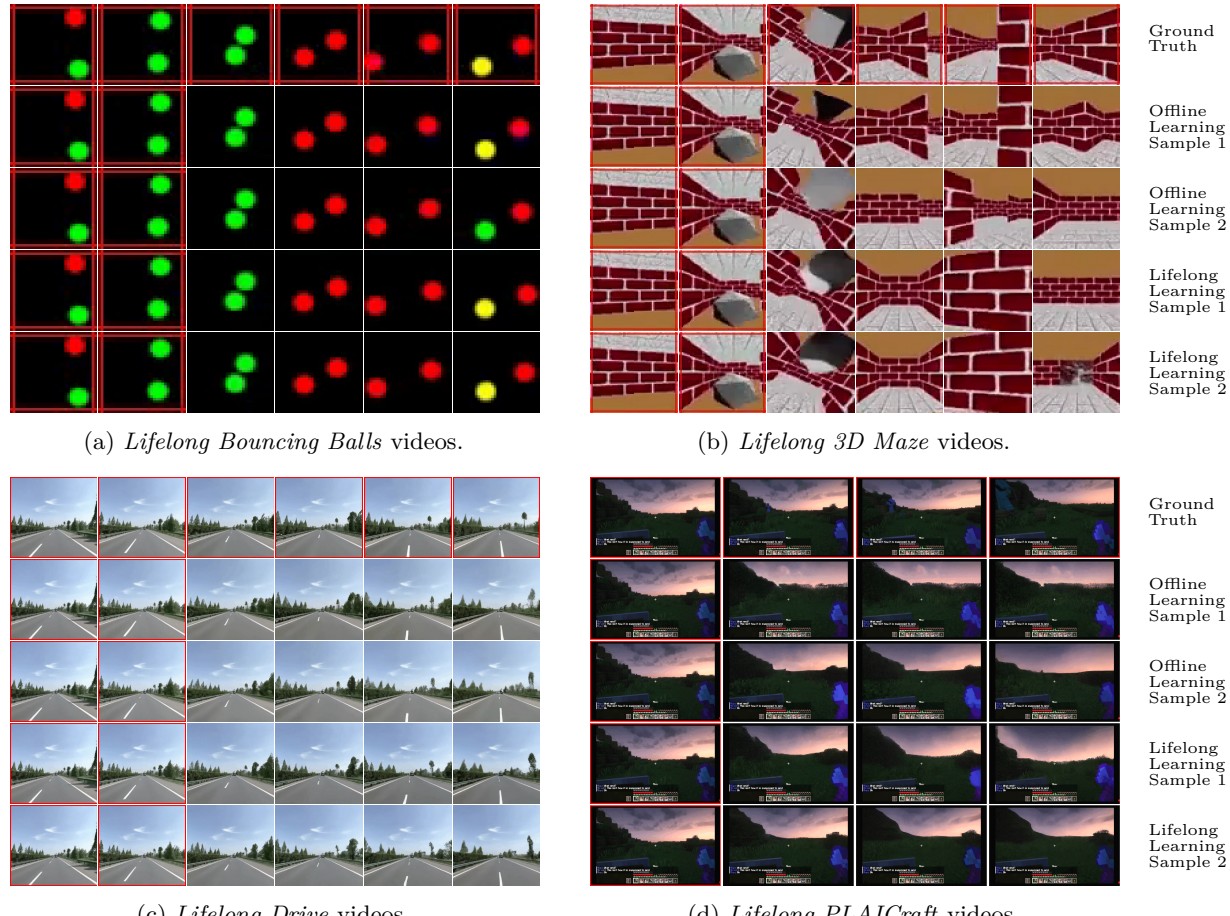

(a) *Lifelong Bouncing Balls* videos.

(b) *Lifelong 3D Maze* videos.

(c) *Lifelong Drive* videos.

(d) *Lifelong PLAICraft* videos.

Figure 1: Ground truth video frames (1st row of each subfigure), offline learned models' generated videos (2nd and 3rd rows), and lifelong learned models' generated video frames (4th and 5th rows) for our datasets. The leftmost columns highlighted in red show the frame conditioned upon the model. Videos generated by lifelong learned models trained with experience replay are diverse, visually plausible, and are comparable to offline models in visual fidelity. Figures are best viewed zoomed in.

These bodies of work suggest a strong desire to avoid non-i.i.d. learning, making it reasonable to assume that lifelong learning of video models from a single autocorrelated video stream might be highly challenging.

The primary contribution of this paper is an empirical proof-of-concept that diffusion-based video models can learn in a lifelong manner from a single autocorrelated video stream while obtaining similar performance to standard (offline) i.i.d. training. We demonstrate successful lifelong learning even on highly nonstationary partially-observable 3D domains across multiple datasets, model architectures, parameter sizes, and optimizers. Remarkably, successful lifelong learning does not require a complex setup. We find that a minimal set of established continual learning techniques, such as experience replay with limited memory, is sufficient to make lifelong- and i.i.d.-trained models qualitatively indistinguishable given the same numbers of gradient steps and batch size.

As a secondary contribution, we introduce four single video lifelong learning datasets with varying levels of temporal correlation, repetitiveness, perceptual complexity, and non-stationarity. Even with an academic computational budget, we observe stable learning and reliable short-range extrapolations on every dataset. As video modeling is a component of world modeling, our findings have the potential to open up new life-like streaming approaches to learning, planning, and control in embodied agents.

## 2 Background

**Video Diffusion Models**   Video Diffusion Models (VDMs) are a class of generative models capable of synthesizing high-quality, temporally consistent videos (Voleti et al., 2022; Ho et al., 2022; Harvey et al., 2022; Höppe et al., 2022; Green et al., 2024; Blattmann et al., 2023b;a; Lu et al., 2024; Brooks et al., 2024). Rooted in the principles of denoising diffusion, video diffusion models progressively refine random noise into coherent video frames across a series of iterative denoising steps. Extending image diffusion models to videos requires capturing temporal dependencies between frames while preserving single-frame quality, which existing work typically achieves using attention mechanisms.

**Lifelong Learning**   The goal of lifelong or continual learning is enabling models to continuously learn from new data with minimal forgetting of what was learned before (De Lange et al., 2022; Wang et al., 2024; van de Ven et al., 2024; Yoo & Wood, 2022). Approaches to lifelong learning include using regularization to penalize changes to parts of the network that encode previously learned information (Kirkpatrick et al., 2017; Zenke et al., 2017; Li & Hoiem, 2017), improving the plasticity or stability of the optimization algorithm (Dohare et al., 2024; Hess et al., 2023; Yoo et al., 2024), and enforcing the encoding of different tasks in orthogonal parts of the model (Rusu et al., 2016; Serra et al., 2018; Zeng et al., 2019). Another popular approach is replay, whereby the model revisits samples representative of past data along with the current training data (Robins, 1995). Such replayed samples can be obtained from generative models (Shin et al., 2017), but often they are sampled from a memory buffer containing past training data, an approach referred to as experience replay (Chaudhry et al., 2019; Rolnick et al., 2019; Buzzega et al., 2020; Arani et al., 2022).

Most work on neural network lifelong learning has focused on the highly simplified problem setting where a classification model is trained on a sequence of non-overlapping tasks, with each task seen only once. Often the model can train on each new task until convergence (referred to as the offline setting), although some works only allow a single pass over the data of each task (the online setting) (Aljundi et al., 2019; Chen et al., 2020). Beyond classification, lifelong learning research has explored the continual training of generative models, including GANs, VAEs and diffusion models for image generation (Zhai et al., 2019; Lesort et al., 2019; Egorov et al., 2021; Smith et al., 2024). However, no work thus far has explored lifelong learning of video diffusion models. The lifelong learning community has also shown interest in moving beyond learning in a strictly task-based manner. To create data streams with more complex temporal correlations, blurry task boundaries (Bang et al., 2022; Moon et al., 2023) and repetition of previously seen concepts (Hemati et al., 2025) have been used. Benchmarks for lifelong learning have also been constructed based on data from autonomous driving (Verwimp et al., 2023), using image datasets collected through time (Bornschein et al., 2023) and by concatenating thousands of short videos (Carreira et al., 2024).

## 3 Datasets

Exploring the possibility of learning video models in a lifelong fashion requires (1) long continuous video streams and (2) varying levels of complexity to gauge challenges and limitations. As discussed in Section 2, existing video datasets usually consist of many short videos that may or may not be temporally related. To that end, we introduce four new video datasets derived from single video streams without semantic discontinuities that vary in perceptual complexity, stochasticity, temporal correlation, and degree of non-stationarity (refer to Appendix C for more details).

**Lifelong Bouncing Balls**   We present two versions of *Lifelong Bouncing Balls*: Versions *O* and *C*.

*Lifelong Bouncing Balls (O)* contains 1 million 32x32 RGB video frames for training (∼28 hours long at 10FPS) and another 1 million video frames for evaluation. The video contains two colored balls that deterministically bounce around in a 2D environment, colliding with boundaries and each other. Upon collision, each ball changes velocity according to the conservation of momentum and cycles through a repeating color sequence of red, yellow, red, and green. Since the ball states are fully observable and the transition dynamics are deterministic, the future frames can be perfectly predicted. Solving *Lifelong Bouncing Balls (O)* requires learning the 2D environment's **deterministic** dynamics and retaining the balls' color transition histories from a **correlated** and **repetitive** video stream.

*Lifelong Bouncing Balls (C)* introduces non-stationarity on top of *Lifelong Bouncing Balls (O)*. While the balls' motion matches that of (O), the blue channel values of all ball colors increase over time at a constant rate (red, yellow, and green ball colors respectively become fuchsia, white, and aqua by the end of the video). The evaluation set contains frames with balls of all previously observed colors. Solving *Lifelong Bouncing Balls (C)* requires learning the **deterministic** ball dynamics and color transitions from a **correlated** video stream with **unrepetitive** details.

**Lifelong 3D Maze**   The *Lifelong 3D Maze* dataset contains 1 million 64x64 RGB video frames for training (∼14 hours long at 20FPS) and 100,000 video frames for evaluation. The video is a first-person view of an agent that navigates a randomly generated 3D maze. Whenever the agent solves a maze, the walls of the solved maze come down and the walls of an unseen maze rise, at which point the agent attempts to solve the new maze. The mazes contain various sparsely appearing objects, including polyhedral gray rocks that flip the agent upside down upon being touched. Since the maze states are partially observable and their transition dynamics are stochastic, the future frames cannot be perfectly predicted given the past frames. Solving *Lifelong 3D Maze* requires learning the first-person sensory inputs associated with navigating a **stochastic** 3D environment and modeling **infrequent events** from a largely **repetitive** and **correlated** video stream.

**Lifelong Drive**   The *Lifelong Drive* dataset contains 1 million 512x512 RGB video frames for training (∼14 hours long at 20FPS) and 100,000 video frames for evaluation. We encode these video frames using the Stable Diffusion perceptual encoder (Rombach et al., 2022) to a shape of 4x64x64. The video is continuous dashcam footage of a vehicle traveling on a series of highways from Chongqing to Shanghai for 1675 kilometers (View, 2023). The drive contains a diverse range of scenery such as mountains, cities, and tunnels, weather conditions such as sunny, cloudy, and rainy weather, and other cars and trucks moving at different speeds. Solving *Lifelong Drive* requires modeling **real world perceptual data** from a single **correlated** and **nonstationary** video stream.

**Lifelong PLAICraft**   The *Lifelong PLAICraft* dataset contains 1 million 1280x768 RGB video frames for training (∼28 hours long at 10FPS) and 500,000 video frames for evaluation. We encode these video frames using the Stable Diffusion perceptual encoder to a shape of 4x160x96. The training video is a first-person view of an anonymous player with an in-game ID of "Alex" engaged in a multiplayer Minecraft survival world (He et al., 2025). The evaluation video is a first-person view of another anonymous player with an in-game ID of "Kyrie" who explores the same world. The videos capture multiple continuous play sessions spanning several months within this shared multiplayer survival world, showcasing various biomes in all three world dimensions, mining, crafting activities, construction, mob fighting, and player-to-player interactions. The Minecraft world contains aspects that repeat (ex. day-night cycle, players visiting their homes) and do not repeat (ex. felled trees, player chat logs). Solving *Lifelong PLAICraft* requires learning a **highly nonstationary** environment that changes in **multiple timescales** from a single **correlated** video stream.

## 4   Model and Training Regime

**Model**   We use video diffusion models with U-Net (Harvey et al., 2022) and Transformer (Lu et al., 2024) neural architectures. The diffusion models' networks operate on $K$ video frames at a time, where $K = 10$ for *Lifelong Bouncing Balls* and *Lifelong PLAICraft* while $K = 20$ for the *Lifelong 3D Maze* and *Lifelong Drive*. The first $K/2$ video frames are model inputs and the next $K/2$ video frames are prediction targets.

During training, given the first $K/2$ uncorrupted video frames and the second $K/2$ video frames corrupted with Gaussian noise, the model regresses the values of the Gaussian noise (Ho et al., 2020). Specifically, let $\boldsymbol{x}$ be a size $K$ window of consecutive video frames, and $F_{\boldsymbol{\theta}}$ be the denoising neural network. The denoising loss function is defined as

$$\ell(\boldsymbol{\theta}, \boldsymbol{x}) = \mathbb{E}_{\boldsymbol{\epsilon}, s} \left\| \boldsymbol{\epsilon} - F_{\boldsymbol{\theta}}(\boldsymbol{x}^{\mathrm{obs}}, \boldsymbol{x}_s^{\mathrm{lat}}, s) \right\|^2, \tag{1}$$

where $\epsilon \sim \mathcal{N}(0, I)$ is a unit normal random vector, and $s \sim \mathcal{U}(1, S)$ is the diffusion noising timestep. The superscripts obs and lat respectively represent the input and predicted part of $\boldsymbol{x}$ i.e., the first and second $K/2$ frames. The subscript $s$ denotes that the video frames are corrupted to the $s$-th diffusion timestep using $\epsilon$ as noise (Ho et al., 2020). During sampling, given the first $K/2$ clean video frames $\boldsymbol{x}^{\mathrm{obs}}$ and the second

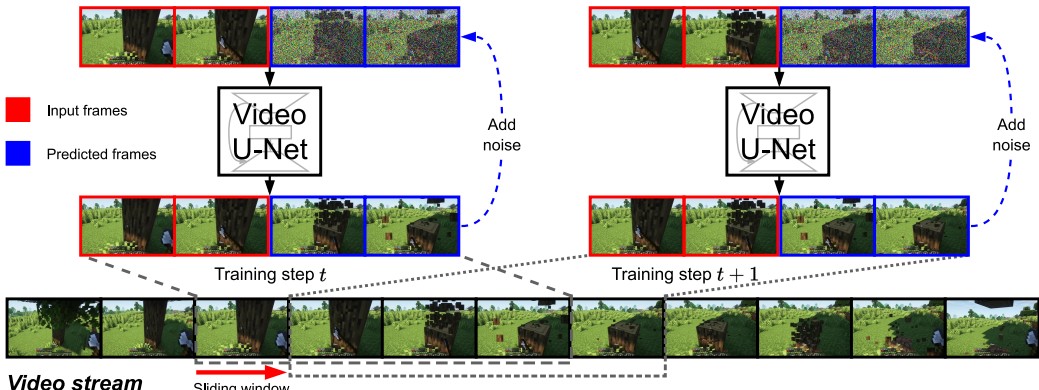

Figure 2: Video diffusion model lifelong learning on a single video stream with $K\!=\!4$. At training step $t$, the model conditions on the frames in the first half of its context window (red) and learns to denoise the frames in the second half of its context window (blue). At training step $t+1$, the model's context window shifts by one video frame, and the same procedure repeats indefinitely.

$K/2$ video frames $\boldsymbol{x}^{\text{lat}}$ filled with Gaussian noise, the model iteratively denoises $\boldsymbol{x}^{\text{lat}}$ to produce a plausible continuation of $\boldsymbol{x}^{\text{obs}}$ using Karras et al. (2022)'s stochastic sampler.

For the rest of this section, we represent the entire video stream as $\mathcal{X}$ and the window of $K$ video frames starting from the $i^{\text{th}}$ frame as $\mathcal{X}^{i:i+K}$. In addition, the expectation in eq. (1) is approximated with a single-sample Monte Carlo estimate.

**Baseline: Offline (i.i.d.) Learning**  As a baseline representative of standard video models, we train models in an *Offline Learning* regime. The $t$-th training step loss is $\mathcal{L}_t^{\text{offline}}(\boldsymbol{\theta}, \mathcal{X}) = \ell(\boldsymbol{\theta}, \mathcal{X}^{i:i+K})$, where $i$ is a uniformly sampled video frame index. The resulting training data resembles i.i.d. segments of $K$ video frames, even though all frames originate from a single autocorrelated stream. When using batch size $N > 1$ (as is common), this i.i.d. sampling is repeated $N$ times.

**Lifelong Learning**  While the Offline Learning regime randomly shuffles the data during training, the Lifelong Learning regime presents data to the model *in order*. This setup reflects the desiderata from Section 1, where the model learns from a real-time data stream. The $t$-th training step loss is $\mathcal{L}_t^{\text{lifelong}}(\boldsymbol{\theta}, \mathcal{X}) = \ell(\boldsymbol{\theta}, \mathcal{X}^{t:t+K})$, i.e., the model receives a sliding window of frames from the single video $\mathcal{X}$. When using batch size $N > 1$, we fill the batch with $N - 1$ copies of the current video $X^{t:t+K}$. This results in a lower variance estimate of eq. (1) by sampling multiple values of $\epsilon$ and $s$.

This Lifelong Learning setup is compatible with numerous techniques proposed by the continual learning community. As a demonstration, we augment the online stream with a replay buffer (Chaudhry et al., 2019) that retains a buffer $\mathcal{M}$ of past video subsequences chosen through reservoir sampling (Vitter, 1985), where the buffer size is chosen to be a small fraction of the total data stream size. With a batch size of $N$, the new $t$-th training step loss is

$$\mathcal{L}_t^{\text{lifelong}}(\boldsymbol{\theta}, \mathcal{X}) = \frac{1}{N}\left(\ell(\boldsymbol{\theta}, \mathcal{X}^{t:t+K}) + \sum_{i=1}^{N-1} \ell(\boldsymbol{\theta}, \mathcal{M}^i)\right), \tag{2}$$

where $\mathcal{M}^i$ is a randomly chosen video from the replay buffer. We refer to this algorithm as *Lifelong Learning* for the rest of this paper. We note that optimizing $\mathcal{L}_t^{\text{lifelong}}$ at each timestep, even with an infinite-size replay buffer, does not produce the same training dynamics as optimizing $\mathcal{L}_t^{\text{offline}}$ since the former scenario only has access to video frames up to time $t$ and the most recent frames are always included in the loss. Lastly, we emphasize that replay is a simple technique that could be replaced with other continual learning methods.

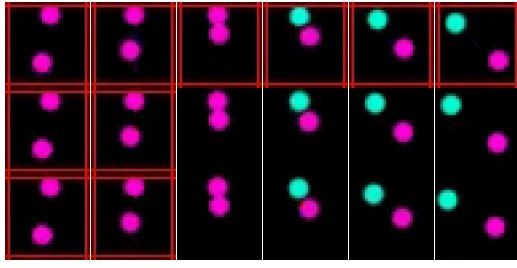

(a) *Lifelong Bouncing Balls (C)* videos.

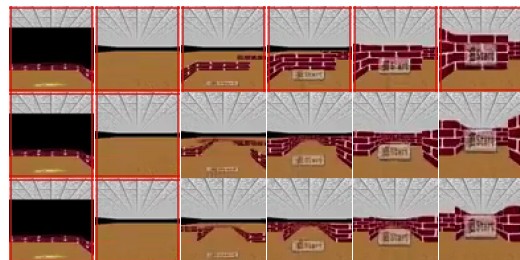

(b) *Lifelong 3D Maze* videos.

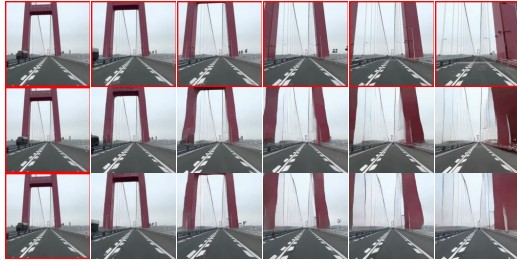

(c) *Lifelong Drive* videos.

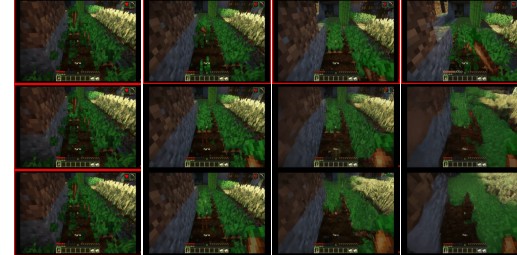

(d) *Lifelong PLAICraft* videos.

Figure 3: Qualitative result comparisons. The top, middle, and bottom rows depict videos from the ground truth data, offline learned model, and lifelong learned model respectively. Additional video samples are presented in the supplementary materials.

## 5 Experiments

This section qualitatively and quantitatively compares offline learned and lifelong learned video diffusion models. The learning algorithms use the same batch size and number of gradient steps, equalizing the amount of computation and runtime memory. We also evaluate two streaming learning baselines, Streaming AdamW and Orthogonal AdamW, which respectively apply the AdamW (Loshchilov & Hutter, 2019) and Orthogonal AdamW (Han et al., 2025) parameter updates to the most recent $K$ frames without replay buffers. We refer the readers to Appendix B for additional experiment details. **Video samples are available in the supplementary materials**.

### 5.1 Lifelong Bouncing Balls

As discussed in Section 3, the *Lifelong Bouncing Balls* datasets feature repetitive and deterministic dynamics, with stationary (Version O) and non-stationary (Version C) color transitions.

**Setup**   The U-Net diffusion models for *Lifelong Bouncing Balls (O)* and *Lifelong Bouncing Balls (C)* respectively have 8 and 74 million parameters, whereas the Transformer diffusion models for both datasets have 42 million parameters. All models are trained with a batch size of 2 and are evaluated using 45 frames that they autoregressively generate conditioned on 5 ground truth frames. Lifelong Learning retains 5 percent of the stream's frames in the replay buffer. We measure the sampled videos' perceptual and temporal coherence using FVD (Unterthiner et al., 2019) and report the loss to measure learning progress. We also measure the models' understanding of the ball movement using minADE (Rasouli, 2020) and of the color transition using our novel ColorKL metric (Appendix A), extracting the ball positions and colors from the frames using 2D convolutions.

**Result**   Qualitative results appear in fig. 1a and fig. 3a. For *Lifelong Bouncing Balls (O)*, Offline Learning and Lifelong Learning produce models that generate visually compelling ball colors and trajectories, correctly handling different ball-to-ball and ball-to-wall collisions. While no model perfectly recovers the deterministic ground truth trajectories, the videos generated by offline and lifelong learned models are not perceptibly different in quality. We observe the same behaviors for *Lifelong Bouncing Balls (C)*. Despite the non-stationary

| Method | Train Stream | | | | Test Stream | | | |
| --- | --- | --- | --- | --- | --- | --- | --- | --- |
| | FVD | Loss$^{\times 10^{-5}}$ | minADE | ColorKL | FVD | Loss$^{\times 10^{-5}}$ | minADE | ColorKL |
| Offline Learning (U) | 4.5 ±0.1 | 6.3 ±0.1 | 1.74 ±0.03 | 0.006 ±0.001 | 4.7 ±0.2 | 6.2 ±0.2 | 1.71 ±0.06 | 0.006 ±0.001 |
| Lifelong Learning (U) | 4.9 ±0.2 | 6.3 ±0.2 | 1.82 ±0.03 | 0.005 ±0.000 | 4.7 ±0.2 | 6.2 ±0.3 | 1.81 ±0.03 | 0.005 ±0.000 |
| Streaming Adam (U) | 4.7 ±0.1 | 6.7 ±0.1 | 1.91 ±0.01 | 0.006 ±0.0 | 5.0 ±0.2 | 6.7 ±0.1 | 1.84 ±0.03 | 0.005 ±0.0 |
| Orthogonal Adam (U) | 5.2 ±0.2 | 6.8 ±0.1 | 1.99 ±0.09 | 0.005 ±0.001 | 4.8 ±0.1 | 6.7 ±0.1 | 1.92 ±0.07 | 0.005 ±0 |
| Offline Learning (T) | 5.4 ±0.2 | 6.8 ±0.3 | 2.25 ±0.12 | 0.006 ±0.001 | 5.1 ±0.2 | 6.7 ±0.3 | 2.16 ±0.12 | 0.006 ±0.001 |
| Lifelong Learning (T) | 5.5 ±0.1 | 6.6 ±0.2 | 2.22 ±0.05 | 0.005 ±0.001 | 5.0 ±0.3 | 6.5 ±0.2 | 2.16 ±0.06 | 0.005 ±0.001 |

Table 1: *Lifelong Bouncing Balls (O)* metrics computed across two training and three sampling seeds. The configurations (U) and (T) denote U-Net and Transformer-based diffusion architecture.

| Method | Train Stream | | | | Test Stream | | | |
| --- | --- | --- | --- | --- | --- | --- | --- | --- |
| | FVD | Loss$^{\times 10^{-5}}$ | minADE | ColorKL | FVD | Loss$^{\times 10^{-5}}$ | minADE | ColorKL |
| Offline Learning (U) | 5.8 ±0.3 | 6.5 ±0.1 | 2.04 ±0.09 | 0.007 ±0.002 | 5.9 ±0.2 | 6.5 ±0.1 | 2.14 ±0.10 | 0.007 ±0.001 |
| Lifelong Learning (U) | 5.0 ±0.1 | 7.4 ±0.1 | 2.03 ±0.00 | 0.005 ±0.001 | 5.7 ±0.2 | 7.5 ±0.1 | 2.06 ±0.00 | 0.005 ±0.000 |
| Streaming Adam (U) | 357.4 ±1.8 | 2240 ±110 | 2.61 ±0.08 | 0.021 ±0.002 | 343.6 ±1.2 | 2252 ±108 | 2.73 ±0.11 | 0.022 ±0.0 |
| Orthogonal Adam (U) | 355.0 ±1.7 | 2327 ±13 | 2.72 ±0.02 | 0.024 ±0.001 | 345.4 ±1.3 | 2301 ±13 | 2.83 ±0.01 | 0.023 ±0.003 |
| Offline Learning (T) | 5.0 ±0.2 | 8.4 ±0.1 | 2.24 ±0.05 | 0.006 ±0.000 | 5.6 ±0.1 | 8.5 ±0.1 | 2.19 ±0.02 | 0.006 ±0.000 |
| Lifelong Learning (T) | 5.3 ±0.1 | 8.1 ±0.0 | 2.29 ±0.03 | 0.006 ±0.001 | 6.4 ±0.3 | 8.1 ±0.0 | 2.27 ±0.03 | 0.006 ±0.000 |

Table 2: *Lifelong Bouncing Balls (C)* metrics computed across two training and three sampling seeds. The configurations (U) and (T) denote U-Net and Transformer-based diffusion architecture.

color changes, both learning algorithms generate realistic ball trajectories and colors, regardless of whether we test on frames from the beginning or the end of the video stream. The quantitative results in table 1 and table 2 mirror the qualitative results. Offline Learning and Lifelong Learning perform similarly on both the stationary *Lifelong Bouncing Balls (O)* and non-stationary *Lifelong Bouncing Balls (C)* datasets. However, Streaming AdamW and Orthogonal AdamW perform comparably to the aforementioned methods on *Lifelong Bouncing Balls (O)* but significantly underperform on *Lifelong Bouncing Balls (C)* from forgetting the earlier ball colors. This suggests that while temporal correlation can be addressed by optimizer-based streaming learning methods, techniques that effectively mitigate forgetting is required to address non-repetitiveness. Overall, the presence of temporal correlation and the degree of non-repetitiveness does not pose a significant learning challenge to Lifelong Learning with replay buffers.

## 5.2 Lifelong 3D Maze

The *Lifelong 3D Maze* dataset is a step up in complexity from *Lifelong Bouncing Balls*. The video stream frames are renderings of a 3D space and feature stochastic dynamics and rare events.

**Setup** The U-Net and Transformer-based diffusion models from this section respectively have 78 million and 110 million parameters. All models are trained with a batch size of 4, of which two elements are the current timestep video frame subsequence, and are evaluated using 40 frames autoregressively generated from conditioning on 10 ground truth frames. Lifelong Learning retains 5 percent of the video stream's frames in the replay buffer. We measure perceptual and temporal coherence using FVD and JEDi (Luo et al., 2025) and learning progress using the diffusion loss.

**Result** Qualitative results appear in fig. 1b and fig. 3b. Both Offline Learning and Lifelong Learning models generate coherent maze trajectories, successfully modeling sparsely occurring event sequences such as camera inversion and the rising of the maze walls. Both models make the occasional mistake, such as deforming polyhedral gray rocks during camera inversion. Quantitative results appear in table 3. We find that the

| Method | Train Stream | | | Test Stream | | |
|---|---|---|---|---|---|---|
| | FVD | JEDi | Loss | FVD | JEDi | Loss |
| Offline Learning (U-Net) | 34.2 $\pm$1.3 | 0.083 $\pm$0.002 | 0.005 $\pm$0.0 | 28.5 $\pm$1.2 | 0.060 $\pm$0.001 | 0.005 $\pm$0.0 |
| Lifelong Learning (U-Net) | 41.2 $\pm$0.4 | 0.073 $\pm$0.006 | 0.006 $\pm$0.0 | 30.7 $\pm$0.1 | 0.061 $\pm$0.003 | 0.006 $\pm$0.0 |
| Streaming Adam (U-Net) | 130.6 $\pm$0.6 | 0.142 $\pm$0.002 | 0.009 $\pm$0.0 | 36.9 $\pm$1.9 | 0.104 $\pm$0.004 | 0.006 $\pm$0.0 |
| Orthogonal Adam (U-Net) | 135.0 $\pm$2.1 | 0.140 $\pm$0.002 | 0.009 $\pm$0.0 | 45.9 $\pm$0.4 | 0.129 $\pm$0.004 | 0.006 $\pm$0.0 |
| Offline Learning (Transformer) | 30.2 $\pm$0.9 | 0.088 $\pm$0.005 | 0.006 $\pm$0.0 | 32.6 $\pm$1.0 | 0.082 $\pm$0.005 | 0.006 $\pm$0.0 |
| Lifelong Learning (Transformer) | 32.1 $\pm$0.9 | 0.082 $\pm$0.005 | 0.006 $\pm$0.0 | 30.2 $\pm$0.8 | 0.070 $\pm$0.001 | 0.006 $\pm$0.0 |

Table 3: *Lifelong 3D Maze* metrics computed across one training and three sampling seeds.

| Method | Train Stream | | | Test Stream | | |
|---|---|---|---|---|---|---|
| | FVD | JEDi | Loss | FVD | JEDi | Loss |
| Offline Learning (U-Net) | 15.3 $\pm$0.2 | 0.071 $\pm$0.000 | 0.029 $\pm$0.0 | 17.9 $\pm$0.3 | 0.102 $\pm$0.001 | 0.026 $\pm$0.0 |
| Lifelong Learning (U-Net) | 17.0 $\pm$0.2 | 0.072 $\pm$0.000 | 0.030 $\pm$0.0 | 21.8 $\pm$0.3 | 0.106 $\pm$0.000 | 0.027 $\pm$0.0 |
| Streaming Adam (U-Net) | 322.8 $\pm$1.2 | 0.608 $\pm$0.002 | 0.041 $\pm$0.0 | 183.4 $\pm$1.7 | 0.343 $\pm$0.000 | 0.029 $\pm$0.0 |
| Orthogonal Adam (U-Net) | 294.4 $\pm$2.3 | 0.590 $\pm$0.001 | 0.041 $\pm$0.0 | 180.0 $\pm$0.7 | 0.315 $\pm$0.0 | 0.029 $\pm$0.0 |
| Offline Learning (Transformer) | 10.4 $\pm$0.2 | 0.023 $\pm$0.000 | 0.027 $\pm$0.0 | 11.2 $\pm$0.1 | 0.036 $\pm$0.000 | 0.024 $\pm$0.0 |
| Lifelong Learning (Transformer) | 10.7 $\pm$0.1 | 0.023 $\pm$0.000 | 0.029 $\pm$0.0 | 12.9 $\pm$0.1 | 0.039 $\pm$0.000 | 0.025 $\pm$0.0 |

Table 4: *Lifelong Drive* metrics computed across one training and three sampling seeds.

Offline and Lifelong Learning metrics are generally similar, and neither methods significantly outperform the other across all metrics, architectures, and stream types. Streaming AdamW and Orthogonal AdamW perform comparably worse, with worse train stream performance than test stream performance due to forgetting from the train stream's mild distribution shift (see appendix C). Overall, we see on qualitative and quantitative fronts that Lifelong Learning performs well and can capture information on rare events when learning from correlated frames generated by a stochastic environment, despite heavy data imbalance between rare events and regular maze traversal frames.

## 5.3 Lifelong Drive

The *Lifelong Drive* dataset features real-world perceptual data captured in a 3D environment similar to *Lifelong 3D Maze*, but with higher frame resolution and greater perceptual complexity.

**Setup**  The U-Net and Transformer-based diffusion models in this section respectively have 80 and 300 million parameters. Lifelong Learning retains 20 percent of the video stream's frames in the replay buffer. All models are trained with a batch size of 8, of which two elements are the current timestep video frame subsequence, and evaluation is performed on videos with 20 frames, 10 of which are model-generated given the preceding 10 frames. We report FVD, JEDi, and model loss.

**Result**  See fig. 1c and fig. 3c for qualitative results. Similar to 3D Maze, Lifelong Learning models produce realistic dynamics of moving in three dimensions. The generated videos from models trained with either algorithm are realistic and are capable of modeling actions like lane changes, though there exists a mild autoregressive drift less present in simpler datasets (ex. fig. 3c bridge shape). Quantitative results appear in table 4. Offline Learning and Lifelong Learning perform similarly across metrics, architectures, and stream types. We see that changing the model architecture and size has a much more significant impact on model performance than the learning algorithm, suggesting that video diffusion models can be effectively lifelong learned from an autocorrelated video stream. The Streaming AdamW and Orthogonal AdamW results again show that learning from an in-order data stream without replay buffers degrades performance.

| | Train Stream | | | Test Stream | | |
|---|---|---|---|---|---|---|
| Method | FVD | JEDi | Loss | FVD | JEDi | Loss |
| Offline Learning (U-Net) | 59.7 $_{\pm 0.2}$ | 0.874 $_{\pm 0.007}$ | 0.034 $_{\pm 0.0}$ | 119.8 $_{\pm 1.3}$ | 1.119 $_{\pm 0.004}$ | 0.042 $_{\pm 0.0}$ |
| Lifelong Learning (U-Net) | 62.9 $_{\pm 0.6}$ | 0.876 $_{\pm 0.003}$ | 0.034 $_{\pm 0.0}$ | 130.8 $_{\pm 1.1}$ | 1.189 $_{\pm 0.003}$ | 0.042 $_{\pm 0.0}$ |
| Streaming Adam (U-Net) | 239.5 $_{\pm 1.5}$ | 2.038 $_{\pm 0.014}$ | 0.047 $_{\pm 0.0}$ | 270.6 $_{\pm 1.9}$ | 2.281 $_{\pm 0.004}$ | 0.050 $_{\pm 0.0}$ |
| Orthogonal Adam (U-Net) | 220.5 $_{\pm 0.8}$ | 1.608 $_{\pm 0.006}$ | 0.048 $_{\pm 0.0}$ | 251.7 $_{\pm 0.8}$ | 1.824 $_{\pm 0.009}$ | 0.052 $_{\pm 0.0}$ |
| Offline Learning (Transformer) | 23.7 $_{\pm 0.1}$ | 0.209 $_{\pm 0.001}$ | 0.024 $_{\pm 0.0}$ | 27.7 $_{\pm 0.3}$ | 0.223 $_{\pm 0.002}$ | 0.024 $_{\pm 0.0}$ |
| Lifelong Learning (Transformer) | 21.9 $_{\pm 0.4}$ | 0.197 $_{\pm 0.003}$ | 0.025 $_{\pm 0.0}$ | 26.3 $_{\pm 0.3}$ | 0.208 $_{\pm 0.001}$ | 0.024 $_{\pm 0.0}$ |

Table 5: *Lifelong PLAICraft* metrics computed across one training and three sampling seeds.

### 5.4 Lifelong PLAICraft

The *Lifelong PLAICraft* dataset is the most challenging in terms of temporal dynamics out of the four datasets. The video stream is continuously recorded from a partially observable environment containing multi-agent interaction and a highly diverse perceptual and action space.

**Setup**   The U-Net and Transformer-based diffusion models in this section respectively have 80 and 300 million parameters. Lifelong Learning retains 20 percent of the video stream's frames in the replay buffer. All models are trained with a batch size of 8, of which two elements are the current timestep video frame subsequence, and evaluation is performed on videos with 10 frames, 5 of which are model-generated conditioned on the preceding 5 frames. We report FVD, JEDi, and model loss.

**Result**   See fig. 1d and fig. 3d for qualitative results. Both Offline Learning and Lifelong Learning models capture perceptual details about Minecraft video frames. Objects present in every gameplay frame (player name, item bar, and the equipped item) are consistently included in all Offline and Lifelong generations, and even chat logs are faithfully rendered (fig. 1d). Frequently observed temporal dynamics such as item movement during walking and flashing of the bottom right recording dot are captured better than player movement and actions. We hypothesize that a larger model size and careful hyperparameter tuning achievable on commercial-scale compute will significantly improve the sample quality for both learning algorithms. Nevertheless, the generated videos from offline and lifelong learned models are not perceptually distinguishable in quality. Quantitative results appear in table 5. On both training and test streams, Offline Learning and Lifelong Learning perform similarly, with neither approach consistently outperforming the other across metrics and architectures. Consistent with the findings from other datasets, Streaming AdamW and Orthogonal AdamW underperform Lifelong Learning. In summary, we find that Lifelong Learning can perform comparably to Offline Learning with minimal hyperparameter tuning, even on videos as complex as *Lifelong PLAICraft*.

### 5.5 Discussion

The reported metrics for Offline and Lifelong Learning across all datasets are remarkably similar. While the confidence intervals for both learning algorithms do not always overlap for every metric, dataset, and model architecture configuration, the performance metric means are largely similar and other control variables such as model architecture, parameter size, and dataset have substantially stronger effects on the metrics. Combined with the qualitative indistinguishability of offline versus lifelong model samples, we conclude that video diffusion models can be effectively lifelong-learned.

**Replay buffer ablation studies.**   We investigate model sensitivity to the replay buffer size for every dataset in fig. 4 and Appendix D. Specifically, we assess the future frame generation capabilities of four different U-Net video diffusion learning configurations: *No Replay* that eschews replay buffers (identical to Streaming AdamW), *Experience Replay* that retains 5–20 percent of the video stream frames (identical to Lifelong Learning), *Full Replay* that retains 100 percent of the video stream frames, and *Offline Learning* that learns from an i.i.d. sampled video segments. We find that storing 5–20 percent of video stream frames, which translates to 5–20 times less data storage than Offline Learning, is often sufficient and that unlimited

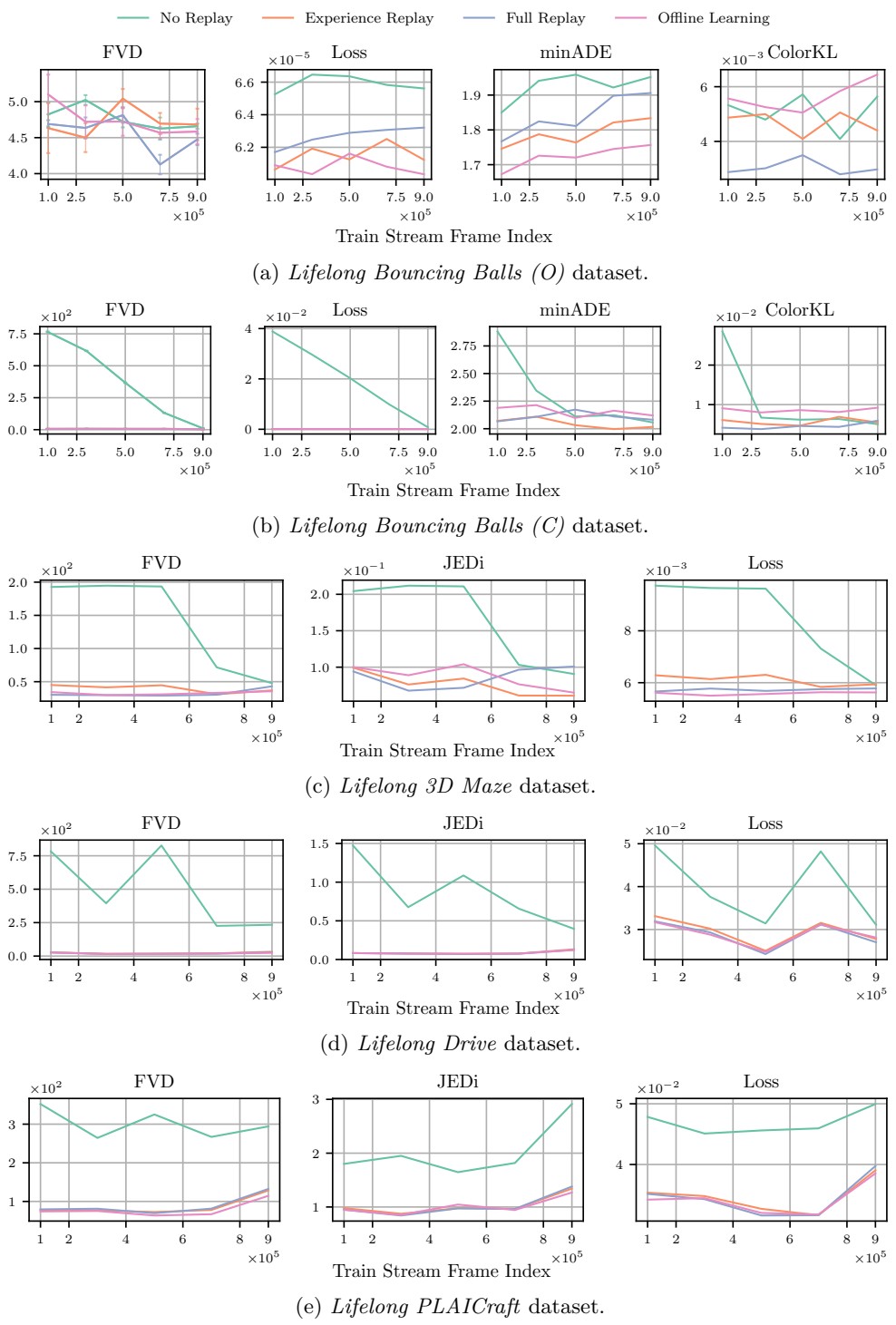

Figure 4: Frame generation performance metrics of fully trained U-Net video diffusion models, computed from frames selected from different points in the training stream. Each point represents evaluation on 1,000 consecutive frame subsequences that follow the corresponding training stream frame index.

replay buffer size does not lead to significantly stronger results for our diverse group of datasets. Finally, we observe that the addition of the replay buffer causes Experience Replay to perform similarly to Offline Learning on video subsequences from different training stream timesteps, suggesting that forgetting is not a

major obstacle given replay buffers. These results indicate that experience replay with a modest buffer size is a strong and simple baseline when lifelong learned video diffusion models are in memory-constrained settings.

**Does learning occur throughout the entire data stream?** To demonstrate that learning realistic models on these datasets is non-trivial, we plot the change in the video diffusion models' test stream quantitative metrics during training for all datasets in fig. 6. Every learning algorithm continues to improve in performance as training progresses, indicating that the generative modeling tasks cannot easily be mastered after the models train on a moderate number of video frames.

**Combining experience replay with Orthogonal AdamW.** Inspired by continual learning methods that combine replay with stabilizing optimizers (Yoo et al., 2024; Urettini & Carta, 2025), we evaluate a lifelong learning approach that minimizes eq. (2)'s replay loss with the Orthogonal AdamW optimizer instead of AdamW and report the results in Appendix E. The resulting models perform comparably to regular experience replay, suggesting that our findings are not highly sensitive to the specifics of optimizer configuration.

## 6 Related Work

Our work is closely related to continual generative modeling (Nguyen et al., 2018; Ramapuram et al., 2020; Masip et al., 2024; Smith et al., 2024; Zając et al., 2023; Campo et al., 2020; Chen et al., 2022), in particular to methods that focus on diffusion models and video generative models. Prior work on diffusion model lifelong learning (Masip et al., 2024; Smith et al., 2024; Zając et al., 2023; Liu et al., 2025) focuses on image modeling under the task-based continual learning setup where data grouped into disjoint tasks arrives in large batches. In contrast, our work focuses on lifelong learning of video diffusion models capable of capturing temporal correlations in video frames by learning from a single video stream. Prior work on video generative model lifelong learning (Campo et al., 2020; Chen et al., 2022) trains VAE models that can generate future frames again in the task-based continual learning setup. Unlike our setup, these methods assume the presence of rigid task boundaries and the ability to train to convergence on large data batches. In contrast, our lifelong learning datasets do not have notions of tasks, and our models only have access to data as they appear in the stream. This setup better reflects the data streams from which biological and future embodied agents learn.

Our work is also related to online learning of sequence processing models (Zucchet et al., 2023; Carreira et al., 2024; Bornschein et al., 2024; Liu et al., 2024; Han et al., 2025). Notably, Carreira et al. (2024); Han et al. (2025) learns discriminative video prediction models from the concatenation of loosely related videos. In contrast, our work learns generative video models from a single autocorrelated stream. Liu et al. (2024) learns a linear regression-based world model in an online fashion, an approach that we note is not amenable to high-dimensional videos.

Lastly, prior work has, like us, introduced video datasets for continual learning. Villa et al. (2022); Tang et al. (2024)'s datasets contain many short videos that can be learned under a task-based continual learning setup. Carreira et al. (2024)'s datasets construct one very long data stream by concatenating multiple short-to-medium length videos, but their data are not publicly available. Singh et al. (2016)'s dataset, while not originally developed for continual learning, contains a series of short-to-medium-length videos of the real world that were collected via Google Glass. We note that their dataset has semantic discontinuities, whereas our datasets come from single, continuous video streams.

## 7 Conclusion

This paper explores lifelong learning of diffusion models from continuous video streams, which has never been investigated to the best of our knowledge. We establish the feasibility of learning video diffusion models from a single autocorrelated video stream in a lifelong fashion. Our lifelong learning approach is simple, using no specialty techniques beyond a minimal replay buffer, and is robust across various data streams, model architectures, parameter sizes, and optimizer configurations. To promote further research into this area, we introduce four single video datasets that test how different video stream characteristics affect lifelong learning of video diffusion models.

The ability to train video diffusion models directly from continuous data streams holds significant promise, particularly for video world model learning in embodied agents (Agarwal et al., 2025; Alonso et al., 2024; Huang et al., 2023; Escontrela et al., 2024) and compute-efficient foundation model adaptation (Smith et al., 2024). Our results suggest that learning generative models from strongly correlated data streams may be less challenging than previously assumed, echoing prior results on discriminative lifelong learning (Bornschein et al., 2024; Carreira et al., 2024; Han et al., 2025). This paper's preliminary investigations lay the groundwork for advancing lifelong learning of video generative models by massively scaling continuous video streams and model sizes, incorporating advanced diffusion architectures, and developing new learning algorithms.

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

## A    The ColorKL Metric

The ColorKL metric measures how faithful a model-generated video's color transition statistics are to the ground truth color transition statistics at the dataset level. It is defined as

$$\frac{1}{|\mathcal{C}|} \sum_{c_{old} \in \mathcal{C}} D_{\mathrm{KL}} \left( p_*(c_{new}|c_{old}) \parallel p_{model}(c_{new}|c_{old}) \right) \tag{3}$$

where $D_{\mathrm{KL}}$ is the KL divergence, $\mathcal{C} = \{red, yellow, green\}$ is the set of possible ball colors, $p_*(c_{new}|c_{old})$ is the ground truth probability for a ball to switch colors from $c_{old}$ to $c_{new}$, and $p_{model}$ is the empirical probability of color transitions from $c_{old}$ to $c_{new}$ in the model-generated video dataset. We note that if the model's context window is long enough to always capture two past ball bounces for all balls, the ball colors can be deterministically predicted. If the model's context is not long enough, as is the case in our experiments, the ground truth transition probabilities $p_*$ state that the balls always transition from yellow to red and green to red but have a 50/50 chance of transitioning from red to yellow or green.

## B    Additional Experiment Details

The minimum average displacement error (minADE) metric is computed by selecting the trajectory with the lowest average displacement error from 3 sampled trajectories for each evaluation video subsequence. The ColorKL metric is computed by tallying the transition statistics from 3 sampled trajectories for each evaluation video subsequence and computing the KL divergence of this empirical distribution with the ground truth transition statistics. The loss metric is calculated by uniformly sampling 10 noise levels for all evaluation set samples per model checkpoint, computing the diffusion MSE loss, and averaging the results.

As computing all metrics on the entirety of the video streams is prohibitively expensive, we select 1,000 video subsequences from the train and test streams and calculate the metrics on those video frames for all reported metrics. All training and sampling seeds compute the metrics on the same set of video subsequences. For datasets without significant changes in video frame details throughout the video stream (*Lifelong Bouncing Balls (O)*), we select the first 1,000 video subsequences from the evaluation video stream. For datasets with significant changes in video frame details throughout the video stream (*Lifelong Bouncing Balls (C), Lifelong Drive, Lifelong PLAICraft*), we evenly select 1,000 video subsequences across the entire evaluation video stream with equal spacing. For *Lifelong 3D Maze*, the train stream evaluation is done across 1,000 video subsequences evenly selected from the stream and the test stream evaluation is done across the first 1,000 video subsequences from the stream due to the subtle train stream perceptual changes detailed in Appendix C.

All models are optimized with AdamW with learning rate 1e-4. The optimizers for U-Net models have weight decay of 1e-5, whereas the optimizers for Transformer models have weight decay of 0 for *Lifelong Bouncing Balls* and 1e-6 for all other experiments. All U-Net models have the *num_res_blocks* hyperparameter set to 1. The *num_channels* hyperparameter is set to 64, 192, 128, 128, 128 for *Lifelong Bouncing Balls (O), Lifelong Bouncing Balls (C), Lifelong 3D Maze, Lifelong Drive,* and *Lifelong PLAICraft* experiments respectively. All VDT models have the *num_layers* hyperparameter set to 12. The *hidden_layer_size* and *patch_size* hyperparameters are set to 384, 384, 640, 1024, 1024 and 2, 2, 4, 4, 8 for *Lifelong Bouncing Balls (O), Lifelong Bouncing Balls (C), Lifelong 3D Maze, Lifelong Drive,* and *Lifelong PLAICraft* experiments respectively. We applied gradient clipping with a threshold of 1 to lifelong learned VDT models for Lifelong PLAICraft and Lifelong Drive experiments to mitigate infrequent but sporadic gradient norm spikes.

The U-Net and VDT experiments for *Lifelong Bouncing Balls* took 3 days to complete on a single GeForce RTX 2080 GPU. The U-Net and VDT experiments for *Lifelong 3D Maze* took 7 days to complete on two Tesla V100 GPUs. The U-Net experiments for *Lifelong Drive* and *Lifelong PLAICraft* took 4 days to complete on 4 A5000 GPUs, and the VDT experiments for *Lifelong Drive* and *Lifelong PLAICraft* took 6 days to complete on 4 L40 GPUs. There was no substantial runtime difference between Offline Learning and Lifelong Learning since both methods employ the same batch sizes and the same number of gradient steps.

## C  Additional Dataset Curation Details

**Lifelong Bouncing Balls**  videos were created by randomly sampling the two balls' initial positions and velocities and deterministically updating them to satisfy the conservation of momentum.

**Lifelong 3D Maze**  was created by concatenating two 10-hour-long Lifelong 3D Maze YouTube videos (Dprotp, 2018; Screensavers, 2020) at the point where a maze from the first video is solved and the maze from the second video begins, as the maze screensaver teleports the agent to a newly generated maze on the completion of the previous maze. This concatenation ensures that there is no sudden switch in the environment dynamics (a slight perceptual switch exists as shown in fig. 5).

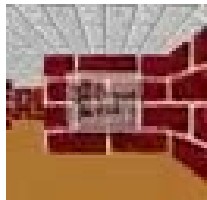

(a) Example frame from video 1.

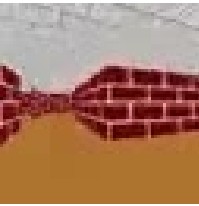

(b) Example frame from video 1.

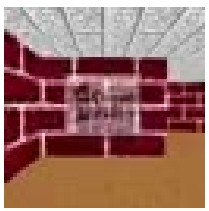

(c) Example frame from video 2.

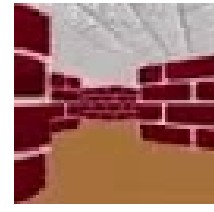

(d) Example frame from video 2.

Figure 5: Example video frames from first and second 10-hour-long YouTube videos used to construct *Lifelong 3D Maze.* While the frames are nearly identical, there are very subtle perceptual differences arising from how the uploaders recorded the two videos.

**Lifelong Drive**  was constructed by first downloading a YouTube video that features a vehicle driving on a series of highways from Chongqing to Shanghai (View, 2023), then cropping and downsampling the video such that its frame dimensions became 512x512. The drive is nearly continuous, with subtle and rare discontinuities occurring when the driver takes breaks in between very long drives. In these scenarios, the video either resumes at the same spot where the vehicle was parked or at a nearby highway location.

**Lifelong PLAICraft**  was constructed by concatenating multiple consecutive gameplay session recordings from the same player of the PLAICraft server (He et al., 2025). All game settings, such as viewing distance, shader configurations, and recording parameters (e.g., resolution and aspect ratio), were consistent across sessions. The dataset includes recordings of two players, Alex and Kyrie, who played in the same survival world alongside hundreds of other players. In their initial gameplay sessions, their starting locations were randomly assigned but could later be altered through travel or teleportation. As in a typical Minecraft survival world, the environment had no boundaries; landscapes were procedurally generated and remained unchanged once created. Player states, including inventories and spawn locations, were preserved between consecutive sessions.

Players' locations and states were generally consistent across most consecutive gameplay sessions. In other words, if a player exited the game at a particular location and state in one session, they would typically resume from the same point in the next session. However, some aspects, such as the player's viewing angle or environment, may be slightly different from reasons such as other players having built structures around the location the player logged off from or other technical reasons related to gameplay recording.

Kyrie's first gameplay session began later than Alex's, and there was some overlap in the locations they visited. Alex generally explored more areas and contributed extensively to building structures, whereas Kyrie primarily played in a village that Alex and other players had previously developed.

## D   Additional Quantitative Results

This section presents the train and test video stream performances of offline training (Offline Learning), lifelong learning using experience replay with limited buffer size (Experience Replay)[2], lifelong learning using experience replay with unlimited buffer size (Full Replay), and lifelong learning without the use of a replay buffer (No Replay) for U-Net models on all datasets.

We make two high-level observations. Firstly, No Replay cannot effectively learn from datasets that are not largely stationary due to forgetting, even on simple datasets like *Lifelong Bouncing Balls (C)*. This suggests that when lifelong learning on video streams that contain non-repeating details, mechanisms that preserve past knowledge are necessary, regardless of how simple the video streams might be. Secondly, we find that Experience Replay and Full Replay performances are not substantially different across the benchmarks, suggesting that storing 5 to 20 percent of video stream frames can be sufficient to have a performant model when lifelong learning video diffusion models.

### D.1   Lifelong Bouncing Balls

| Method | Train Stream | | | | Test Stream | | | |
|---|---|---|---|---|---|---|---|---|
| | FVD | Loss$^{\times 10^{-5}}$ | minADE | ColorKL | FVD | Loss$^{\times 10^{-5}}$ | minADE | ColorKL |
| Offline Learning | 4.5 $\pm 0.1$ | 6.3 $\pm 0.1$ | 1.74 $\pm 0.03$ | 0.006 $\pm 0.001$ | 4.7 $\pm 0.2$ | 6.2 $\pm 0.2$ | 1.71 $\pm 0.06$ | 0.006 $\pm 0.001$ |
| No Replay | 4.7 $\pm 0.1$ | 6.7 $\pm 0.1$ | 1.91 $\pm 0.01$ | 0.006 $\pm 0.0$ | 5.0 $\pm 0.2$ | 6.7 $\pm 0.1$ | 1.84 $\pm 0.03$ | 0.005 $\pm 0.0$ |
| Experience Replay | 4.9 $\pm 0.2$ | 6.3 $\pm 0.2$ | 1.82 $\pm 0.03$ | 0.005 $\pm 0.000$ | 4.7 $\pm 0.2$ | 6.2 $\pm 0.3$ | 1.81 $\pm 0.03$ | 0.005 $\pm 0.000$ |
| Full Replay | 4.7 $\pm 0.1$ | 6.3 $\pm 0.1$ | 1.80 $\pm 0.01$ | 0.004 $\pm 0.001$ | 4.6 $\pm 0.1$ | 6.2 $\pm 0.0$ | 1.76 $\pm 0.02$ | 0.003 $\pm 0.001$ |

Table 6: *Lifelong Bouncing Balls (O)* performance metrics for U-Net models. The left and right columns respectively denote training and test video stream results computed across two training and three sampling random seeds.

| Method | Train Stream | | | | Test Stream | | | |
|---|---|---|---|---|---|---|---|---|
| | FVD | Loss$^{\times 10^{-5}}$ | minADE | ColorKL | FVD | Loss$^{\times 10^{-5}}$ | minADE | ColorKL |
| Offline Learning | 5.8 $\pm 0.3$ | 6.5 $\pm 0.1$ | 2.04 $\pm 0.09$ | 0.007 $\pm 0.002$ | 5.9 $\pm 0.2$ | 6.5 $\pm 0.1$ | 2.14 $\pm 0.10$ | 0.007 $\pm 0.001$ |
| No Replay | 357.4 $\pm 1.8$ | 2240 $\pm 110$ | 2.61 $\pm 0.08$ | 0.021 $\pm 0.002$ | 343.6 $\pm 1.2$ | 2252 $\pm 108$ | 2.73 $\pm 0.11$ | 0.022 $\pm 0.0$ |
| Experience Replay | 5.0 $\pm 0.1$ | 7.4 $\pm 0.1$ | 2.03 $\pm 0.00$ | 0.005 $\pm 0.001$ | 5.7 $\pm 0.2$ | 7.5 $\pm 0.1$ | 2.06 $\pm 0.00$ | 0.005 $\pm 0.000$ |
| Full Replay | 5.0 $\pm 0.1$ | 8.0 $\pm 0.1$ | 2.08 $\pm 0.03$ | 0.005 $\pm 0.001$ | 5.6 $\pm 0.2$ | 7.9 $\pm 0.2$ | 2.12 $\pm 0.0$ | 0.006 $\pm 0.001$ |

Table 7: *Lifelong Bouncing Balls (C)* performance metrics for U-Net models. The left and right columns respectively denote training and test video stream results computed across two training and three sampling random seeds.

### D.2   Lifelong 3D Maze

### D.3   Lifelong Drive

### D.4   Lifelong PLAICraft

---

[2]This configuration is referred to as Lifelong Learning in the main text's section 5.

| Method | Train Stream | | | Test Stream | | |
|---|---|---|---|---|---|---|
| | FVD | JEDi | Loss | FVD | JEDi | Loss |
| Offline Learning | 34.2 ±1.3 | 0.083 ±0.002 | 0.005 ±0.0 | 28.5 ±1.2 | 0.060 ±0.001 | 0.005 ±0.0 |
| No Replay | 130.6 ±0.6 | 0.142 ±0.002 | 0.009 ±0.0 | 36.9 ±1.9 | 0.104 ±0.004 | 0.006 ±0.0 |
| Experience Replay | 41.2 ±0.4 | 0.073 ±0.006 | 0.006 ±0.0 | 30.7 ±0.1 | 0.061 ±0.003 | 0.006 ±0.0 |
| Full Replay | 34.0 ±0.3 | 0.080 ±0.002 | 0.006 ±0.0 | 35.2 ±0.5 | 0.075 ±0.006 | 0.006 ±0.0 |

Table 8: *Lifelong 3D Maze* metrics for experience replay with different replay buffer sizes for U-Net models. The metrics are computed across one training and three sampling seeds.

| Method | Train Stream | | | Test Stream | | |
|---|---|---|---|---|---|---|
| | FVD | JEDi | Loss | FVD | JEDi | Loss |
| Offline Learning | 15.3 ±0.2 | 0.071 ±0.000 | 0.029 ±0.0 | 17.9 ±0.3 | 0.102 ±0.001 | 0.026 ±0.0 |
| No Replay | 322.8 ±1.2 | 0.608 ±0.002 | 0.041 ±0.0 | 183.4 ±1.7 | 0.343 ±0.000 | 0.029 ±0.0 |
| Experience Replay | 17.0 ±0.2 | 0.072 ±0.000 | 0.030 ±0.0 | 21.8 ±0.3 | 0.106 ±0.000 | 0.027 ±0.0 |
| Full Replay | 15.8 ±0.0 | 0.072 ±0.000 | 0.030 ±0.0 | 19.3 ±0.2 | 0.096 ±0.000 | 0.026 ±0.0 |

Table 9: *Lifelong Drive* metrics for experience replay with different replay buffer sizes for U-Net models. The metrics are computed across one training and three sampling seeds.

| Method | Train Stream | | | Test Stream | | |
|---|---|---|---|---|---|---|
| | FVD | JEDi | Loss | FVD | JEDi | Loss |
| Offline Learning | 59.7 ±0.2 | 0.874 ±0.007 | 0.034 ±0.0 | 119.8 ±1.3 | 1.119 ±0.004 | 0.042 ±0.0 |
| No Replay | 239.5 ±1.5 | 2.038 ±0.014 | 0.047 ±0.0 | 270.6 ±1.9 | 2.281 ±0.004 | 0.050 ±0.0 |
| Experience Replay | 62.9 ±0.6 | 0.876 ±0.003 | 0.034 ±0.0 | 130.8 ±1.1 | 1.189 ±0.003 | 0.042 ±0.0 |
| Full Replay | 67.8 ±0.7 | 0.867 ±0.004 | 0.034 ±0.0 | 131.6 ±0.7 | 1.170 ±0.001 | 0.042 ±0.0 |

Table 10: *Lifelong PLAICraft* metrics for experience replay with different replay buffer sizes for U-Net models. The metrics are computed across one training and three sampling seeds.

# E Orthogonal Replay

We denote the AdamW-optimized experience replay objective as Experience Replay, and denote the Orthogonal-AdamW optimized experience replay objective as Orthogonal Replay.

| Method | Train Stream | | | | Test Stream | | | |
|---|---|---|---|---|---|---|---|---|
| | FVD | Loss$^{\times 10^{-5}}$ | minADE | ColorKL | FVD | Loss$^{\times 10^{-5}}$ | minADE | ColorKL |
| Experience Replay (U) | 4.9 ±0.2 | 6.3 ±0.2 | 1.82 ±0.03 | 0.005 ±0.000 | 4.7 ±0.2 | 6.2 ±0.3 | 1.81 ±0.03 | 0.005 ±0.000 |
| Orthogonal Replay (U) | 4.8 ±0.1 | 6.4 ±0.0 | 1.82 ±0.02 | 0.006 ±0.000 | 4.9 ±0.3 | 6.2 ±0.0 | 1.75 ±0.00 | 0.005 ±0.000 |
| Experience Replay (T) | 5.5 ±0.1 | 6.6 ±0.2 | 2.22 ±0.05 | 0.005 ±0.001 | 5.0 ±0.3 | 6.5 ±0.2 | 2.16 ±0.06 | 0.005 ±0.001 |
| Orthogonal Replay (T) | 5.5 ±0.2 | 6.4 ±0.2 | 2.17 ±0.02 | 0.006 ±0.002 | 5.3 ±0.1 | 6.3 ±0.2 | 2.11 ±0.01 | 0.007 ±0.001 |

Table 11: *Lifelong Bouncing Balls (O)* metrics computed across two training and three sampling seeds. The configurations (U) and (T) denote U-Net and Transformer-based diffusion architecture.

| Method | Train Stream | | | | Test Stream | | | |
|---|---|---|---|---|---|---|---|---|
| | FVD | Loss$^{\times 10^{-5}}$ | minADE | ColorKL | FVD | Loss$^{\times 10^{-5}}$ | minADE | ColorKL |
| Experience Replay (U) | 5.0 ±0.1 | 7.4 ±0.1 | 2.03 ±0.00 | 0.005 ±0.001 | 5.7 ±0.2 | 7.5 ±0.1 | 2.06 ±0.00 | 0.005 ±0.000 |
| Orthogonal Replay (U) | 5.2 ±0.1 | 8.2 ±0.1 | 2.10 ±0.11 | 0.004 ±0.001 | 6.0 ±0.1 | 8.2 ±0.1 | 2.12 ±0.07 | 0.004 ±0.000 |
| Experience Replay (T) | 5.3 ±0.1 | 8.1 ±0.0 | 2.29 ±0.03 | 0.006 ±0.001 | 6.4 ±0.3 | 8.1 ±0.0 | 2.27 ±0.03 | 0.006 ±0.000 |
| Orthogonal Replay (T) | 5.4 ±0.2 | 7.7 ±0.1 | 2.23 ±0.01 | 0.006 ±0.001 | 6.0 ±0.2 | 7.8 ±0.1 | 2.25 ±0.00 | 0.006 ±0.001 |

Table 12: *Lifelong Bouncing Balls (C)* metrics computed across two training and three sampling seeds. The configurations (U) and (T) denote U-Net and Transformer-based diffusion architecture.

| Method | Train Stream | | | Test Stream | | |
|---|---|---|---|---|---|---|
| | FVD | JEDi | Loss | FVD | JEDi | Loss |
| Experience Replay (U-Net) | 41.2 ±0.4 | 0.073 ±0.006 | 0.006 ±0.0 | 30.7 ±0.1 | 0.061 ±0.003 | 0.006 ±0.0 |
| Orthogonal Replay (U-Net) | 41.3 ±0.4 | 0.080 ±0.003 | 0.006 ±0.0 | 31.8 ±0.7 | 0.084 ±0.001 | 0.006 ±0.0 |
| Experience Replay (Transformer) | 32.1 ±0.9 | 0.082 ±0.005 | 0.006 ±0.0 | 30.2 ±0.8 | 0.070 ±0.001 | 0.006 ±0.0 |
| Orthogonal Replay (Transformer) | 34.5 ±0.6 | 0.067 ±0.003 | 0.006 ±0.0 | 29.7 ±1.2 | 0.071 ±0.002 | 0.006 ±0.0 |

Table 13: *Lifelong 3D Maze* metrics computed across one training and three sampling seeds.

| Method | Train Stream | | | Test Stream | | |
|---|---|---|---|---|---|---|
| | FVD | JEDi | Loss | FVD | JEDi | Loss |
| Experience Replay (U-Net) | 17.0 ±0.2 | 0.072 ±0.000 | 0.030 ±0.0 | 21.8 ±0.3 | 0.106 ±0.000 | 0.027 ±0.0 |
| Orthogonal Replay (U-Net) | 18.8 ±0.1 | 0.078 ±0.000 | 0.030 ±0.0 | 25.6 ±0.2 | 0.125 ±0.000 | 0.026 ±0.0 |
| Experience Replay (Transformer) | 10.7 ±0.1 | 0.023 ±0.000 | 0.029 ±0.0 | 12.9 ±0.1 | 0.039 ±0.000 | 0.025 ±0.0 |
| Orthogonal Replay (Transformer) | 11.1 ±0.1 | 0.027 ±0.000 | 0.029 ±0.0 | 13.6 ±0.2 | 0.044 ±0.000 | 0.025 ±0.0 |

Table 14: *Lifelong Drive* metrics computed across one training and three sampling seeds.

| Method | Train Stream | | | Test Stream | | |
|---|---|---|---|---|---|---|
| | FVD | JEDi | Loss | FVD | JEDi | Loss |
| Experience Replay (U-Net) | 62.9 ±0.6 | 0.876 ±0.003 | 0.034 ±0.0 | 130.8 ±1.1 | 1.189 ±0.003 | 0.042 ±0.0 |
| Orthogonal Replay (U-Net) | 69.2 ±1.1 | 0.842 ±0.006 | 0.035 ±0.0 | 136.3 ±0.9 | 1.206 ±0.001 | 0.043 ±0.0 |
| Experience Replay (Transformer) | 21.9 ±0.4 | 0.197 ±0.003 | 0.025 ±0.0 | 26.3 ±0.3 | 0.208 ±0.001 | 0.024 ±0.0 |
| Orthogonal Replay (Transformer) | 21.9 ±0.1 | 0.182 ±0.002 | 0.024 ±0.0 | 26.0 ±0.3 | 0.192 ±0.001 | 0.024 ±0.0 |

Table 15: *Lifelong PLAICraft* metrics computed across one training and three sampling seeds.

# F    Optimization Time Performance Metrics

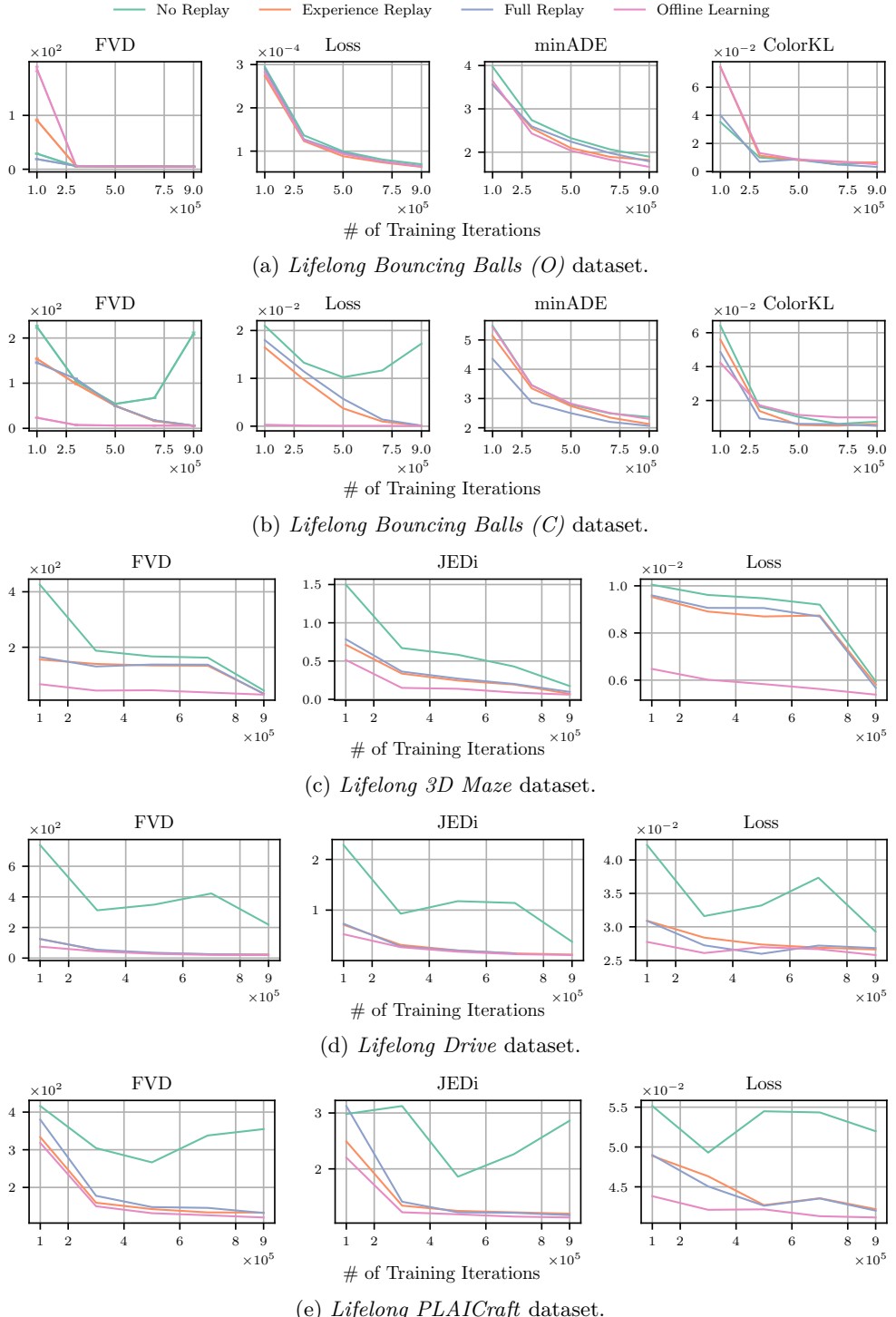

Figure 6: Test stream performance metrics for the U-Net model checkpoints at different training iterations (refer to Appendix D for baseline details). The plots show the improvement in model quality as online training progresses. All models generally improve the longer they are trained.

