# OpenReview forum: "Lifelong Learning of Video Diffusion Models From a Single Video Stream"
_TMLR — Rejected by TMLR_

### Review · Reviewer_nYaE · 2025-11-08

**Summary Of Contributions:**

The paper demonstrates that autoregressive video diffusion models can be trained continually from a single autocorrelated stream and achieve performance comparable to offline i.i.d. training when optimization budgets are matched. The method uses standard windowed diffusion training with an experience-replay buffer that stores a small fraction of past subsequences. The authors also introduce four single-stream datasets, namely Bouncing Balls, 3D Maze, Drive, and PLAICraft, each with one million frames. Experiments with U-Net and Transformer architectures evaluate FCD, JEDi, and loss on fixed subsequence sets, finding lifelong+replay closely tracks offline while streaming optimizers whithout replay degrade.

**Audience:**

Yes

**Audience Explanation:**

The work targets a central continual-learning scenario with a single-stream and autocorrelated video, and it delivers an empirical proof that strong generative models can learn in this regime with a minimal replay recipe. These findings should interest readers evaluating trade-offs between memory, data ordering, and compute in practical online systems.

**Claims And Evidence:**

Yes

**Claims Explanation:**

Evidence is provided across multiple datasets and architectures. Evaluations uses fixed subsequence sets and equal gradient-step budgets for fair comparisons. Buffer ablations conclude that retaining roughly 5-20% of frames is often sufficient and much larger buffers yield little benefit, reinforcing the claim's practicality.

**Requested Changes:**

- I would suggest an offline baseline limited to the same number of unique windows retained by the replay buffer at each memory fraction to isolate storage vs ordering effect.
- I suggest consolidate resource reporting into one table: time, memory, and buffer.
- Small suggestion on the terminology consistency of "lifelong/continual/streaming/online" and "AdamW/Adam" to avoid confusion.

---

> ### Author Response · Authors · 2025-11-29
>
> We thank the reviewer for noting the interestingness of our problem setting and for recognizing the strength of our evidence. We address your points in detail below.
>
> >  I would suggest an offline baseline limited to the same number of unique windows retained by the replay buffer at each memory fraction to isolate storage vs ordering effect.
>
> Thank you for the interesting suggestion. We agree it would be valuable to more clearly disentangle storage and ordering effects. However, restricting the offline baseline to training only on the windows retained in the replay buffer does not fully disentangle storage and ordering effects, because experience replay still observes every window in the training stream at least once, regardless of whether it is stored. This causes experience replay to be exposed to more data compared to the offline baseline.
>
> We have examined storage and ordering effects through other analyses already included in the paper. In particular, we studied how experience replay with different buffer sizes compares to the offline baseline (Figure 4; Appendix D) and how in-order versus shuffled training affects performance over optimization (Appendix F). We hope these analyses address the reviewer’s concern from complementary angles.
>
> > I suggest consolidating resource reporting into one table: time, memory, and buffer.
>
> We will add an appendix section that consolidates the metrics from Section 5, Appendix D, and Appendix F into a single table for each dataset in the camera-ready version.
>
> > Small suggestion on the terminology consistency of "lifelong/continual/streaming/online" and "AdamW/Adam" to avoid confusion.
>
> We have done this in the revised draft.

---

### Review · Reviewer_jxYM · 2025-11-17

**Summary Of Contributions:**

This paper is the first to explore training autoregressive video diffusion models directly from a continuous lifelong video stream. Traditional video diffusion models rely on randomly shuffled mini-batches from large, static datasets, which require substantial storage and often ignore temporal continuity. In contrast, this work introduces four novel single-stream video datasets designed specifically for lifelong learning settings. The authors demonstrate that video diffusion models trained in this lifelong setting—augmented only with a experience replay buffer—can match the performance of offline models trained on i.i.d. data.

**Audience:**

Yes

**Audience Explanation:**

This paper is likely to attract certain interest within the TMLR audience, as it lays foundational work for lifelong generative modeling. The contributions align well with the interests of both generative modeling researchers, who are exploring scalable and adaptive generative architectures, and applied ML engineers in robotics and world modeling, who require data-efficient systems capable of learning continuously from streaming sensory input.

**Claims And Evidence:**

No

**Claims Explanation:**

The paper makes two primary claims: (1) the proposed datasets provide a meaningful testbed for future research on lifelong learning, and (2) a simple replay buffer is sufficient to achieve robust lifelong learning of video diffusion models, reaching performance comparable to offline-trained models.

The first claim is well-supported. The visualizations and supplementary materials clearly demonstrate that the datasets—particularly Drive and PLAICraft—exhibit substantial diversity, non-stationarity, and long-range temporal autocorrelation. These characteristics make them compelling and challenging benchmarks for lifelong video modeling.

However, the second claim is only partially supported.
The evaluation is limited by two factors:

- **Short prediction horizons**: The models predict only very short futures (≈0.5 seconds). This horizon is insufficient to conclusively demonstrate robustness to compounding errors or long-range temporal dynamics—both central challenges in lifelong generative modeling and world modeling.

- **Large replay buffers**: The replay buffer sizes used for Drive and PLAICraft (20% of the full stream, corresponding to ~2.8 and ~5.6 hours of video) are extremely large by continual learning standards. With buffers this large, a significant fraction of the underlying distribution is retained, which weakens the claim that the models are truly learning under memory constraints. In contrast, prior continual learning work on replay buffers[1] (including the papers cited by the authors) typically uses very small buffers (often <1%, or on the order of ~100 samples). Such setups more realistically reflect the constraints of embodied agents, where storage is severely limited.

Given this, the current results do not fully validate the claim that replay alone is a robust and memory-efficient solution for lifelong learning. To strengthen the argument—especially for embodied-agent use cases—the authors should evaluate substantially smaller replay buffers or demonstrate performance–memory tradeoffs more thoroughly.

[1] Chaudhry, Arslan, et al. "On tiny episodic memories in continual learning." arXiv preprint arXiv:1902.10486 (2019).

**Requested Changes:**

I recommend revising the current claim that "lifelong learning of video diffusion models is solved by replay buffer" since it is not accurate. While the experiments show that a large replay buffer (up to 20% of the data stream) can match offline performance, this setup does not convincingly demonstrate that replay alone solves lifelong learning, especially in realistic settings with tight memory constraints.

A more impactful direction would be to characterize the limits of replay:

- How small can the buffer be before performance degrades sharply?
- Are some datasets more sensitive to buffer size than others?

A buffer that stores 20% of the stream (~3–6 hours of video in this case) covers too much of the data distribution to be considered a realistic “lifelong” constraint, especially in embodied or on-device settings. Demonstrating the performance trade-offs with smaller replay buffers would make the work significantly more insightful and generalize better to real-world continual learning scenarios.

---

> ### Author Response · Authors · 2025-11-29
>
> We thank the reviewer for recognizing the strength of our datasets to be a meaningful testbed for future lifelong learning research, as well as the interestingness of our findings. We address your points in detail below.
>
> > The evaluation is limited by short horizontal horizons: The models predict only very short futures (~0.5 seconds).
>
> All qualitative results in the paper, as well as all supplementary material contents, are sourced from videos that are 3 to 5 seconds long. These videos were autoregressively generated by conditioning on the previously generated frames for four to five times the size of the model’s context length for all datasets. We find that the lifelong and offline models’ video samples under this regime are qualitatively indistinguishable. We will highlight this fact in the main paper for the camera-ready draft.
>
> For quantitative results, metrics for Lifelong Bouncing Balls and Lifelong 3D Maze were calculated on autoregressively sampled videos that are 5 seconds long, since sampling these lower-resolution videos is relatively cheap. Lifelong Drive and Lifelong PLAICraft metrics were calculated on a single generation pass (0.5 seconds) since sampling these higher-resolution videos is more expensive. To address the reviewer’s concerns, **we present new rebuttal results** that compare offline learning and experience replay (Lifelong Learning) for U-Net models trained on Lifelong Drive and Lifelong PLAICraft datasets on generated videos that are respectively 5 and 2.5 seconds long.
>
> | Dataset | Method | Train FVD | Train JEDi | Train Loss | Test FVD | Test JEDi | Test Loss |
> |-|-|-|-|-|-|-|-|
> | Lifelong Drive | Lifelong Learning | 41.2 | 0.184 | 0.030 | 61.8 | 0.270 | 0.026 |
> | Lifelong Drive | Offline Learning | 36.5 | 0.183 | 0.030 | 42.5 | 0.282 | 0.026 |
> | Lifelong PLAICraft | Lifelong Learning | 217.6 | 1.392 | 0.034 | 276.4 | 1.497 | 0.042 |
> | Lifelong PLAICraft | Offline Learning | 250.6 | 1.374 | 0.034 | 333.0 | 1.462 | 0.042 |
>
> Similar to short-range extrapolation results, we neither Lifelong Learning nor Offline Learning dominate the other on all metrics. We hypothesize that the FVD gap between the two methods is wider for these results compared to short-range extrapolation results because the Inception network behind FVD is more sensitive to autoregressive error buildup compared to the V-JEPA network behind JEDi. Overall, these results further reinforce our claim that lifelong and offline learning can perform comparably.
>
>
> > The evaluation is limited by large replay buffers
> > - A buffer that stores 20% of the stream (~3–6 hours of video in this case) covers too much of the data distribution to be considered a realistic “lifelong” constraint, especially in embodied or on-device settings.
> > - While the experiments show that a large replay buffer can match offline performance, this setup does not convincingly demonstrate that replay alone solves lifelong learning under tight memory constraints.
>
> Lifelong learning methods need not be maximally memory-efficient to be valuable, and can also focus on other aspects of learning, such as computational efficiency [1]. Our approach enables training generative models on non-i.i.d. video streams in a way that matches the training compute and performance of the i.i.d. baseline, while storing 5–20x less data. This tradeoff is valuable, especially in practical contexts such as edge devices that already support storing larger data volumes. For example, the Inspire 3 drone [2] contains a 1TB hard drive capable of storing 20 days' worth of 1080p videos [3]. Experience replay with larger buffer sizes is thus plausible for real-world deployed lifelong learned edge devices.

---

> ### Author Response · Authors · 2025-11-29
>
> > How small can the buffer be before performance degrades sharply?
>
> To address the reviewer’s questions, **we present new rebuttal results** for U-Net video diffusion models trained with experience replay that stores 5% of the data stream for Lifelong Drive and Lifelong PLAICraft.
>
> | Dataset | Method | Train FVD | Train JEDi | Train Loss | Test FVD | Test JEDi | Test Loss |
> |-|-|-|-|-|-|-|-|
> | Lifelong Drive | 0% Replay | 322.8 | 0.608 | 0.041 | 183.4 | 0.343 | 0.029 |
> | Lifelong Drive | 5% Replay | 29.3 | 0.131 | 0.039 | 33.2 | 0.166 | 0.028 |
> | Lifelong Drive | 20% Replay | 17.0 | 0.072 | 0.030 | 21.8 | 0.106 | 0.027 |
> | Lifelong PLAICraft | 0% Replay | 239.5 | 2.038 | 0.047 | 270.6 | 2.280 | 0.050 |
> | Lifelong PLAICraft | 5% Replay | 69.8 | 0.842 | 0.036 | 142.4 | 1.200 | 0.045 |
> | Lifelong PLAICraft | 20% Replay | 62.9 | 0.876 | 0.034 | 130.8 | 1.189 | 0.042 |
>
>
> We find that while there is a general performance dropoff compared to the 20% memory experience replay, the dropoff is minor compared to that of the 0% memory experience replay setup. This suggests that, owing to dataset complexity, the replay buffer size where additional capacity does not significantly improve the model performance is larger for Lifelong Drive and Lifelong PLAICraft than for Lifelong Bouncing Balls and Lifelong 3D Maze.
>
> > Are some datasets more sensitive to buffer size than others?
>
> Our original results in Figure 4 and Appendix D established that the performance degradation associated with replay buffer presence is much less pronounced for loosely stationary datasets (Lifelong Bouncing Balls (O), Lifelong 3D Maze) compared to non-stationary datasets (Lifelong Bouncing Balls (C), Lifelong Drive, Lifelong PLAICraft).
>
> > I recommend revising the current claim that "lifelong learning of video diffusion models is solved by replay buffer".
>
> We note that our claim in the abstract is “lifelong learning of video diffusion models can also be as effective as standard offline training given the same number of gradient steps, and that this main result can be achieved using experience replay that retain a subset of the preceding video stream” rather than “lifelong learning of video diffusion models is solved by replay buffer”. As you point out, there is much room for improvement in learning video generative models from a single data stream, such as improving memory efficiency. Our preliminary investigations lay the groundwork for these advancements.
>
> ### References
>
> [1] Verwimp, E., Aljundi, R., Ben-David, S., Bethge, M., Cossu, A., Gepperth, A., ... & Van de Ven, G. M. (2023). Continual learning: Applications and the road forward. arXiv preprint arXiv:2311.11908.
>
> [2] DJI. (n.d.). DJI Inspire 3 – Master the unseen. https://www.dji.com/inspire-3
>
> [3] Canale, A. (2021, September 27). The Growing Size of Media: Just How Much Information Can Be Stored on 1TB? Infosecurity Magazine. https://www.infosecurity-magazine.com/blogs/how-much-information-stored-1tb/

---

> > ### Comment · Reviewer_jxYM · 2025-12-02
> >
> > I thank the authors for the detailed clarification and the inclusion of new experimental results.
> > Based on the rebuttal, I understand that:
> > - The replay buffer is effective specifically for longer prediction horizons ($2.5 \sim 5$ seconds).
> > - The relationship between buffer size and performance has been analyzed.
> > - Approximately 20% memory retention is required to match the performance of offline learning.
> >
> > However, I still have significant concerns regarding the positioning of the paper and the practical applicability of the proposed method:
> > - **1. Replay Buffer and Efficiency**
> >   - Computational vs. Memory Efficiency: If the argument favors computational efficiency over memory efficiency, the rationale for equalizing the computation cost in the experiments is unclear. This experimental design choice does not appear to be fully supported by the narrative in the current manuscript. Could the authors clarify this inconsistency?
> >   - Edge Device Constraints: The authors argue that edge devices possess large memory capacities (e.g., 1TB, capable of storing 20 days of 1080p video). If such ample storage is assumed, it raises a fundamental question: Why is Online Continual Learning necessary? With that much capacity, one could simply perform Offline Learning directly on the edge device.
> >   - Dataset Scale: Furthermore, relying on only several hours of video seems insufficient for a practical Continual Learning setup by current standards of edge devices (storing 20 days of 1080p video).
> >
> > - **2. Clarity of Claims"Retaining a Subset":**
> >   - The current phrasing—"retain a subset of the preceding video stream"—is potentially misleading. In the context of Continual Learning/Lifelong Learning, readers often interpret "subset" as a negligible fraction (e.g., $<1\%$ or $<100$ samples).
> >   - Explicit Quantities: Since this method requires a 5-20% buffer size to be effective, this significant requirement must be explicitly stated in the abstract and conclusion to avoid overclaiming. The distinction between this requirement and standards of continual learning is substantial.

---

> > > ### Author Response · Authors · 2025-12-05
> > >
> > > We thank the reviewer for acknowledging our rebuttal results, which further support our claim for longer prediction horizons and analyze the relationship between buffer size and performance. We address your remaining concerns in detail below.
> > >
> > > > Computational vs. Memory Efficiency: If the argument favors computational efficiency over memory efficiency, the rationale for equalizing the computation cost in the experiments is unclear. This experimental design choice does not appear to be fully supported by the narrative in the current manuscript. Could the authors clarify this inconsistency?
> > >
> > > Our position is that both compute and memory efficiency are important considerations for single-stream lifelong learning, and that a learning approach need not be maximally memory-efficient to be valuable. However, we do not claim a computational efficiency advantage over offline training with full access to the data. Instead, our advantage lies in being able to learn effectively under autocorrelated, non-stationary streaming data without having access to the entire stream at all points of training, while achieving higher memory efficiency (5–20x less storage) and matching the compute of offline training.
> > >
> > > > Edge Device Constraints: If ample storage (e.g., 1TB, capable of storing 20 days of 1080p video) is assumed, it raises a fundamental question: Why is Online Continual Learning necessary? With that much capacity, one could simply perform Offline Learning directly on the edge device.
> > >
> > > As noted above, the key advantage of lifelong learning in a streaming setting is that models can adapt continuously to a non-stationary, non-i.i.d. data stream. In contrast, a purely offline approach for such streams requires periodic full retraining with i.i.d. sampling from all past frames, which is computationally and memory-wise costly and prevents rapid adaptation. Our replay-based lifelong learning setup avoids this limitation while matching the compute and performance of the offline baseline and using 5–20× less storage.
> > >
> > > > Dataset Scale: Furthermore, relying on only several hours of video seems insufficient for a practical Continual Learning setup by current standards of edge devices (storing 20 days of 1080p video).
> > >
> > > While our current datasets span up to 28 hours of continuous single-sensor videos—substantially longer than the seconds- to minutes-long videos typically used for video model evaluation—we agree that scaling to multi-week video streams is an important direction for future work. One of the paper’s main objectives is to establish the first set of long datasets for continuous, unbroken single-stream video modeling, and we hope this stimulates the development of even longer single-sensor video datasets.
> > >
> > > > Clarity of Claims “Retaining a Subset”: Readers often interpret “subset” as a negligible fraction of the data stream. Since this method requires a 5-20% buffer size to be effective, this significant requirement must be explicitly stated in the abstract and conclusion to avoid overclaiming.
> > >
> > > We thank the reviewer for pointing out the potential ambiguity around the term “subset.” To address this, we will amend the second sentence of the abstract to: “Our work further reveals that this main result can be achieved using experience replay methods that retain only 5–20% of the preceding video stream.” We will also clarify in the conclusion that improving memory efficiency is an important direction for future work. For context, we note that buffer sizes larger than 1% are common in online continual learning evaluations (e.g., up to 10% [1,2,3]).
> > >
> > > We appreciate the reviewer’s thoughtful engagement with our rebuttal and hope that our responses address all outstanding concerns. We would be happy to clarify any remaining questions.
> > >
> > > ### References
> > >
> > > [1] Buzzega, P., Boschini, M., Porrello, A., Abati, D., & Calderara, S. (2020). Dark experience for general continual learning: A strong, simple baseline. In Advances in Neural Information Processing Systems (Vol. 33, pp. 15920–15930).
> > >
> > > [2] Liang, Y.-S., & Li, W.-J. (2023). Loss decoupling for task-agnostic continual learning. In Advances in Neural Information Processing Systems (Vol. 36). https://openreview.net/forum?id=9Oi3YxIBSa
> > >
> > > [3] Yoo, J., Liu, Y., Wood, F., & Pleiss, G. (2024). Layerwise proximal replay: A proximal point method for online continual learning. In Proceedings of the 41st International Conference on Machine Learning (Vol. 235, pp. 57199–57216). https://proceedings.mlr.press/v235/yoo24a.html

---

### Review · Reviewer_9D1N · 2025-11-25

**Summary Of Contributions:**

This paper investigates the feasibility of training autoregressive video diffusion models directly from a single, continuous, autocorrelated video stream. The authors empirically demonstrate that standard video diffusion models (both U-Net and Transformer architectures) can be trained effectively in this lifelong learning setting without complex continual learning algorithms. The key finding is that a simple Experience Replay mechanism, utilizing reservoir sampling to retain a modest subset (5–20%) of the data stream, is sufficient to achieve performance comparable to models trained on offline, i.i.d. shuffled datasets. This paper also introduces four new datasets for streaming lifelong generative video modeling, ranging from simple 2D physics environments (Lifelong Bouncing Balls) to complex, non-stationary 3D environments (Lifelong 3D Maze, Lifelong Drive, and Lifelong PLAICraft/Minecraft). The work establishes a baseline for future research into video-based world models that learn online.

**Audience:**

Yes

**Audience Explanation:**

The findings of this paper are likely to be of some interest to researchers in both generative modeling and continual learning. With the rising prominence of video diffusion models, the ability to train these systems on continuous data streams rather than static, curated datasets addresses a timely and practical problem. Simultaneously, the work offers insights for the continual learning community by empirically demonstrating that simple experience replay can effectively mitigate forgetting in deep generative models trained on highly correlated data, thereby challenging the assumption that such settings require complex, specialized algorithms.

**Broader Impact Concerns:**

I do not have any significant concerns regarding the ethical implications of this work.

**Claims And Evidence:**

Yes

**Claims Explanation:**

The submission provides robust empirical evidence to support its central claim: that video diffusion models can be effectively trained on single, autocorrelated video streams using experience replay. The authors validate their approach across four newly introduced datasets that cover a spectrum of complexity, from simple 2D physics (Bouncing Balls) to complex, non-stationary 3D environments (Minecraft/PLAICraft). This breadth ensures the findings are not specific to a single domain. Additionally, this paper provides a clear side-by-side comparison between the proposed "Lifelong Learning" method and the "Offline (i.i.d.)" upper bound. The quantitative metrics (FVD, JEDi, Loss) and qualitative visualizations (Figure 1 & 3) show that the performance gap is negligible.

**Requested Changes:**

1. The current experimental setup defines the replay buffer size as a percentage (5–20%) of the total video stream length. This implies that memory requirements grow linearly with time, which deviates from the strict lifelong learning assumption of a fixed memory budget and is unrealistic for real-world deployment (e.g., video streams lasting longer than a year). The authors should demonstrate the impact of a fixed buffer size on the model's final performance.

2. The paper only compares "Experience Replay" against "No Replay" baselines, lacking comparisons (or at least a discussion) regarding other standard Continual Learning methods.

3. The paper states that for a batch size of N, the batch is composed of the current data plus N-1 historical samples. Could this structure potentially cause the model to overfit to the most recent data?

4. The algorithmic novelty is limited; the work reads more like an analytical study than a contribution of a new algorithm. Given that the performance of lifelong learning is shown to be very close to that of offline learning, what is the practical significance of this work? What are the concrete or potential real-world applications for the proposed algorithms?

---

> ### Author Response · Authors · 2025-11-29
>
> We thank the reviewer for recognizing both the strength of our empirical evidence and the relevance of our insights for continual learning—specifically, that deep generative models can be effectively trained on highly correlated data without complex algorithms. We address your comments in detail below.
>
> > The current experimental setup defines the replay buffer size as a percentage (5–20%) of the total video stream length. The author should demonstrate the impact of a fixed buffer size on the model’s final performance.
>
> All of our experiments were run with a fixed buffer size corresponding to 5-20% of the total video stream length (1 million frames for all datasets). Therefore, they show the impact of a fixed buffer size.
>
> > The paper only compares "Experience Replay" against "No Replay" baselines, lacking comparisons (or at least a discussion) regarding other standard Continual Learning methods.
>
> We note that all tables in the paper also contain the results for Orthogonal AdamW [1], an optimization-based streaming learning technique developed for streaming learning of discriminative models. We find that Orthogonal AdamW significantly underperforms compared to experience replay on non-stationary video streams. This is unsurprising since the main motivation of Orthogonal AdamW is stabilizing the streaming training dynamics rather than addressing catastrophic forgetting.
>
> Furthermore, the existing papers most relevant to our streaming video generative modelling setup are those that concern a) task-based offline continual learning of video generative models [2,3] and b) streaming video discriminative model training [1,4]. Methods from the former group cannot be applied in our setting since it lacks discrete task boundaries. This is why we baselined the latest work from the second group, Orthogonal AdamW [1], for our experiments.
>
> > The paper states that for a batch size of N, the batch is composed of the current data plus N-1 historical samples. Could this structure potentially cause the model to overfit to the most recent data?
>
> As the reviewer noted, prior work [5] highlighted that the experience replay gradient can be biased toward newer data. To investigate this point, we plotted the performances of fully trained offline trained models and experience replay models for video frames at different video stream timesteps for all datasets in Figure 4. We found that offline learning and experience replay perform comparably on video frames at different video stream timesteps, which suggests that the effect of newer data bias on the model quality is not pronounced.
>
> > The algorithmic novelty is limited; the work reads more like an analytical study than a contribution of a new algorithm. Given that the performance of lifelong learning is shown to be very close to that of offline learning, what is the practical significance of this work? What are the concrete or potential real-world applications for the proposed algorithms?
>
> Our work is best interpreted as a proof of concept that demonstrates the feasibility of efficiently training video generative models on single, unbroken video streams through our datasets and analysis, rather than as a primarily methodological work. Regarding practical significance, video modelling is a core component of world models, and our results represent an important step toward vision-based agents that learn continuously from streaming experience—more akin to how biological agents learn over a lifetime in their environment.
>
> Concretely, this line of work opens the door to embodied agents whose generative world models are themselves learned online, enabling real-time adaptation. Potential applications include self-driving systems that learn from their continuous video streams to anticipate diverse future trajectories (related to Lifelong Drive) and multi-agent or open-ended environments where agents must predict multiple plausible futures as conditions evolve (related to Lifelong PLAICraft).

---

> > ### Author Response · Authors · 2025-11-29
> > **References**
> >
> > [1] Han, T., Gokay, D., Heyward, J., Zhang, C., Zoran, D., Patraucean, V., Carreira, J., Damen, D., & Zisserman, A. (2025). Learning from Streaming Video with Orthogonal Gradients. In Proceedings of the IEEE/CVF Conference on Computer Vision and Pattern Recognition (CVPR) (pp. 13651-13660).
> >
> > [2] Campo, D., Slavic, G., Baydoun, M., Marcenaro, L., & Regazzoni, C. (2020, October). Continual learning of predictive models in video sequences via variational autoencoders. In 2020 IEEE International Conference on Image Processing (ICIP) (pp. 753-757). IEEE.
> >
> > [3] Chen, G., Zhang, W., Lu, H., Gao, S., Wang, Y., Long, M., & Yang, X. (2022). Continual predictive learning from videos. In Proceedings of the IEEE/CVF Conference on Computer Vision and Pattern Recognition (pp. 10728-10737).
> >
> > [4] Carreira, J., King, M., Patraucean, V., Gokay, D., Ionescu, C., Yang, Y., ... & Zisserman, A. (2024). Learning from one continuous video stream. In Proceedings of the IEEE/CVF Conference on Computer Vision and Pattern Recognition (pp. 28751-28761).
> >
> > [5] Caccia, L., Aljundi, R., Asadi, N., Tuytelaars, T., Pineau, J., & Belilovsky, E. (2022). New insights on reducing abrupt representation change in online continual learning. In Proceedings of the International Conference on Learning Representations (ICLR).

---

### Review · Reviewer_QeqN · 2025-11-26

**Summary Of Contributions:**

This paper studies lifelong (streaming) learning for video diffusion models. Instead of assuming access to a large shuffled i.i.d. dataset of short video clips, the authors consider a setting where the model observes a single long video stream in temporal order and must learn a generative model from that stream.
Concretely, they work with fixed-length clips of K frames taken from the stream. Each clip is split into the first K/2 frames: used as conditioning input (the “past”), the last K/2 frames: treated as the future segment to be generated.
They then train a conditional video diffusion model: the future segment is noised using the standard diffusion forward process, and the model is trained to predict the added noise, conditioned on the past frames and the current diffusion timestep. So, while the loss is defined in terms of noise prediction (as in standard diffusion), the semantics are that the model learns the conditional distribution of future frames given past frames.
To support this setting, the authors introduce four custom single-stream video benchmarks. It mostly covers all variations from slow deterministic frames to fast varying dynamic frames.

On the modeling side, the paper does not propose new architectures. It uses existing video diffusion models (a U-Net-based model and a Transformer-based Video Diffusion Transformer, VDT) with a standard ε-prediction diffusion loss. The key difference between training regimes is how clips are sampled:
* Offline Learning: clips are sampled i.i.d. from the full stream (shuffled), mimicking standard offline training.
* Lifelong Learning (their main regime): clips arrive in temporal order as the stream progresses. At each step, the current clip is combined with clips drawn from a replay buffer that stores a small fraction of past clips, maintained via reservoir sampling.
* Streaming baselines (Streaming Adam / Orthogonal Adam): the model trains only on the current clip at each time step, without any replay buffer.
The central empirical claim is that Lifelong Learning with a replay buffer performs similarly to Offline Learning, whereas the streaming baselines without replay often fail, especially on non-stationary streams. On simple, stationary synthetic data (e.g., basic bouncing balls), all methods perform reasonably well and the gap between regimes is small. On more challenging non-stationary or complex streams, Streaming-only training degrades severely, while Lifelong + replay typically stays much closer to the Offline baseline, though usually still slightly worse.

**Audience:**

No

**Audience Explanation:**

Topic of lifelong learning for video models is well researched in computer vision domain, this paper feels like an incomplete slice of a larger project. It offers some empirical observations but lacks the breadth of experiments, analytical reasoning, and intuitive insight needed to turn those observations into a compelling and meaningful contribution.

**Claims And Evidence:**

No

**Claims Explanation:**

There are several concerns and limitations:
1. Scope of comparison: The comparisons are restricted to:
    - Offline vs Lifelong + replay vs Streaming-only,
    - on the authors’ own datasets,
    - using relatively small U-Net and Transformer models. There is no comparison to modern, large pretrained video or autoregressive/diffusion models that are fine-tuned on the stream, which would be a natural and strong baseline in current practice.
2. Use of only custom benchmarks: Because all datasets are constructed by the authors and are not standard benchmarks, it is difficult to judge how the conclusions transfer to more realistic or diverse video data. The setting is informative as a controlled benchmark, but the external validity of the results is limited. The benchmarks are not highlighted which causes more confusion. Refer to table 1 and table 2
- Methodological novelty: The core “lifelong learning” method is simply experience replay applied to a standard conditional diffusion model.
-  No new loss, no new architectural mechanism, and no new continual-learning techniques are introduced.
-  The replay buffer uses uniform reservoir sampling without more advanced strategies. As a result, the contribution is primarily experimental/benchmark-oriented, rather than algorithmic or theoretical.
3. Analysis depth: The paper does not provide a deeper analytical or theoretical explanation of why the streaming + replay regime behaves as it does, or why a performance gap remains between Lifelong and Offline regimes on more complex datasets.
    - Metrics such as FVD, JEDi, minADE, and ColorKL are defined, but the interpretation and discussion of the numerical differences could be clearer and more emphasized.
    - There is no downstream evaluation (e.g., using the learned models as world models for control tasks) to show the practical impact of the observed performance gaps.
4. Positioning within lifelong / continual learning literature: While the paper situates itself as lifelong learning for video diffusion, it does not fully connect to broader continual-learning approaches (e.g., other replay strategies, regularization-based methods, or theoretical perspectives). Nor does it convincingly argue why training from scratch on a single stream is preferable to, or competitive with, approaches that adapt large pretrained models.

**Requested Changes:**

1. Add comparisons on standard benchmarks
   Please evaluate the proposed lifelong/streaming training setup on at least one or two standard video benchmarks in addition to the custom streams. This would help clarify how the method behaves in more widely used settings and make the results easier to interpret and compare.

2. Clarify and highlight the main results in Tables 1 and 2
   The key takeaways from Tables 1 and 2 are not immediately clear. It would help to:
   - more explicitly highlight the main conclusions in the caption and text (e.g., bolding or annotating key numbers),
   - clearly state what constitutes a “good” vs “bad” value for each metric,
   - and summarize, in words, what these tables show about Offline vs Lifelong vs Streaming training.

3. Strengthen analytical / intuitive reasoning
   The paper would benefit from a clearer explanation of why the proposed lifelong setup behaves the way it does.

4. Improve clarity and organization of the writing
   Several sections (especially the experimental setup and the description of the training regimes) are hard to follow on a first read. A clearer, more structured presentation—possibly with a small schematic of the training pipeline and a more concise description of models, losses, and sampling procedures—would make the paper much easier to understand.

---

> ### Author Response · Authors · 2025-11-29
>
> We thank the reviewer for acknowledging the breadth of our dataset design and for clearly outlining their concerns. We address each point in detail below.
>
> > The comparisons of offline vs lifelong learning are restricted to experience replay and streaming baseline, on the author’s own datasets, using relatively small U-Net and transformer models, but not large pre-trained video diffusion models that are fine-tuned.
>
> We thank the reviewer for the interesting suggestion. While we agree these baselines are valuable, we believe they fall outside the scope of this initial, foundational study for two main reasons.
>
> 1. Practical constraints of the motivating deployment setting:
> Our work is motivated by embodied agents and edge-device systems that continuously update their future prediction model from their own visual sensor streams. These platforms can impose strict memory and compute constraints, making it challenging to run or update large pre-trained diffusion models with billions of parameters in real time. For these scenarios, thoroughly investigating the behaviors of small- to medium-scale models (8-300 million parameters) trained from scratch is a necessary first step toward practical lifelong learning.
>
> 2. Scientific value of investigating core lifelong learning dynamics without pre-training biases:
> Our study provides the first systematic investigation of training autoregressive video generative models directly and continuously from a single, autocorrelated video stream. This learning regime is arguably the closest existing analogue to the pre-training free single-stream learning regime of biological agents—unlike prior work that studies video generative continual learning in task-based continual learning settings [1,2]. We perform a comprehensive investigation of video diffusion model lifelong learning in this regime, analyzing the effects of data-stream properties, neural architectures, model sizes, and optimizer design on the model’s final and optimization-time performances.
>
> > It is difficult to judge how the conclusions transfer to more realistic or diverse video data because all datasets are constructed by the authors. Please evaluate the proposed lifelong/streaming training setup on at least one or two standard video benchmarks in addition to the custom streams.
>
> We would like to clarify that our dataset suite already includes driving footage and Minecraft gameplay, which are domains that have been used to evaluate video diffusion models (e.g., [3,4]). Our video streams exhibit heterogeneous perceptual characteristics and complex environment dynamics. Their quality and comprehensiveness have also been acknowledged by reviewers jxYM and 9D1N, and you can examine their video snippets in the supplementary material.
>
> Our primary motivation for constructing 1M-frame single-stream datasets is to enable research into training video models on very long, autocorrelated video streams without discontinuities, which the standard video modelling benchmarks that typically consist of short, shuffled clips do not currently support. To better understand and address the reviewer’s concern, could the reviewer clarify which specific aspects of standard benchmarks they believe are essential for evaluating lifelong, single-stream learning, and why the conclusions drawn from our video streams might not generalize to other video data?
>
> > The core “lifelong learning” method is experience replay applied to a conditional diffusion model, and as a result, the contribution is primarily experimental/benchmark-oriented.
>
> We agree that our work is positioned more as a foundational empirical study than as a methodological contribution. While our approach builds on experience replay, our goal is to establish and analyze the lifelong single-stream video generative model learning setting itself, which, to our knowledge, has not been systematically explored. We also respectfully note that scientific contributions are not limited to new algorithms: applying existing techniques in novel, challenging regimes can reveal new insights essential for future methodological advances. Our datasets and analysis are intended to provide precisely this groundwork.
>
> > Positioning within lifelong / continual learning literature: While the paper situates itself as lifelong learning for video diffusion, it does not fully connect to broader continual-learning approaches.
>
> While we have provided a high-level overview of continual learning methodologies in Section 2 and connections between our problem settings and existing lifelong video modeling work in the related work section (Section 6), we will further expand this to other methodological work in the camera-ready draft.

---

> > ### Comment · Reviewer_QeqN · 2025-12-01
> > **followup on author's concern part 2**
> >
> > ---
> >
> > ### 4. Final recommendations for strengthening the work
> >
> > Summarizing my concrete suggestions:
> >
> > 1. **Compare with existing generative methods on the same data.**
> >
> >     Include at least one baseline from prior streaming/continual generative work (e.g., a GAN or VAE with replay, or the most relevant autoregressive model from *Learning from One Continuous Video Stream*) trained on your single-stream datasets.
> >
> > 2. **Evaluate on a standard streaming benchmark.**
> >
> >     For example, apply your lifelong diffusion setup to a subset of **Ego4D-Stream**, which has already been used to study autoregressive generative models in a continuous video regime. It is not necessary to run every baseline there, but seeing how your approach behaves on this dataset would make the results much more transferable.
> >
> > 3. **Add human-centric or stronger perceptual evaluation.**
> >
> >     Given the limitations of FVD/JEDi and the visible artifacts in some samples, consider a small human study or additional perceptual metrics to better assess qualitative differences between Offline and Lifelong training.
> >
> > 4. **Provide more in-depth analysis of why the method behaves as it does.**
> >
> >     For example, analyze how replay buffer size, non-stationarity level, and model capacity interact; or borrow some analytical tools / diagnostics from Carreira et al.’s *Learning from One Continuous Video Stream* while maintaining the diffusion-specific focus.
> >
> >
> > With these additions, I believe the work would make a much clearer and more compelling contribution.

---

> ### Author Response · Authors · 2025-11-29
>
> > TMLR’s audience will not be interested in the findings of the paper since topics of lifelong learning for video models are well researched in the computer vision domain, and this paper feels like an incomplete slice of a larger project. The paper offers some empirical observations but lacks the breadth of experiments, analytical reasoning, and intuitive insight.
>
> We appreciate the reviewer’s perspective but respectfully disagree with the characterization of the work as incomplete or overly narrow. To our knowledge, this paper presents the first systematic study of video generative model learning from a single video stream, a setting highly relevant to lifelong learning for embodied agents, and introduces four datasets to promote future research.
>
> Our experiments span multiple video streams, architectures, parameter scales, optimizers, and training regimes, and our datasets were carefully designed to individually test key lifelong-learning challenges such as frame repetition, rare events, perceptual space, and environment dynamics complexity. This breadth allows us to evaluate and analyze model behavior under diverse and realistic conditions.
>
> Finally, our empirical findings yield a clear insight: video diffusion models can be trained online from a single stream while achieving performance comparable to offline training, indicating a practical and scalable path for continual video generation. We hope this clarifies the scope and significance of the contribution.
>
>
> > Strengthen analytical / intuitive / theoretical reasoning: The paper would benefit from a clearer explanation of why the proposed lifelong setup behaves the way it does, or why a performance gap remains between Lifelong and Offline regimes on more complex datasets.
>
> We highlight that it is challenging to precisely and theoretically identify why a learning algorithm attains its specific performance in deep learning research, a phenomenon that is further exacerbated when learning from complex non-iid data streams. However, we establish that the intuition that experience replay’s catastrophic forgetting mitigation through repeated data exposure is the cause behind its good performance, which is significantly better than that of purely optimization-based lifelong learning approaches such as Orthogonal AdamW [5]. We attribute this to the latter methods’ primary focus on data stream autocorrelation, which our results on Lifelong Bouncing Balls and Lifelong 3D Mazes demonstrate is not the most significant challenge for lifelong single-stream generative model learning.
>
> Regarding Lifelong and Offline Learning performance gap, we note that neither methods dominate the other in performance across every metric/model/stream pairing for any dataset. This is in contrast to Streaming AdamW and Orthogonal AdamW, whose performance gap to Offline Learning significantly widens the more non-stationary a dataset is.
>
> > Writing clarity and formatting: Concise description of models, losses, and sampling procedures, summarize in words what the tables show on Offline vs Lifelong Learning, benchmarks are not highlighted, highlight what constitutes a good vs bad value for each metric, etc.
>
> We have made formatting adjustments (bolding key sentences, upper case \Cref, etc) and will further improve methodology descriptions for the camera-ready draft. Regarding summarizing in words what the tables show about Offline vs Lifelong Learning, we note that we summarize the tables’ metrics in words in Sections 5.1/5.2/5.3/5.4 and discuss how the training dynamics are affected in Section 5.5.
>
> ### References
>
> [1] Campo, D., Slavic, G., Baydoun, M., Marcenaro, L., & Regazzoni, C. (2020, October). Continual learning of predictive models in video sequences via variational autoencoders. In 2020 IEEE International Conference on Image Processing (ICIP) (pp. 753-757). IEEE.
>
> [2] Chen, G., Zhang, W., Lu, H., Gao, S., Wang, Y., Long, M., & Yang, X. (2022). Continual predictive learning from videos. In Proceedings of the IEEE/CVF Conference on Computer Vision and Pattern Recognition (pp. 10728-10737).
>
> [3] Harvey, W., Naderiparizi, S., Masrani, V., Weilbach, C., & Wood, F. (2022). Flexible diffusion modeling of long videos. Advances in neural information processing systems, 35, 27953-27965.
>
> [4] Ren, X., Lu, Y., Cao, T., Gao, R., Huang, S., Sabour, A., ... & Ling, H. (2025). Cosmos-Drive-Dreams: Scalable Synthetic Driving Data Generation with World Foundation Models. arXiv preprint arXiv:2506.09042.
>
> [5] Han, T., Gokay, D., Heyward, J., Zhang, C., Zoran, D., Patraucean, V., Carreira, J., Damen, D., & Zisserman, A. (2025). Learning from Streaming Video with Orthogonal Gradients. In Proceedings of the IEEE/CVF Conference on Computer Vision and Pattern Recognition (CVPR) (pp. 13651-13660).

---

> ### Comment · Reviewer_QeqN · 2025-12-01
> **Followup on author's concern**
>
> I thank the authors for their detailed responses and for clarifying the intended scope of the work. The additional context is helpful, and I do apologize for unclear request in my prior comments.
>
> ---
>
> ### 1. Scope, pretraining, and prior generative work
>
> The authors emphasize that the goal is a “foundational empirical study” of small– to medium–scale video diffusion models trained **from scratch** on a single continuous stream, motivated by embedded / edge settings where large pretrained models are not practical. I understand this framing. It would help, however, if the paper gave a more concrete definition of the **edge systems** it targets, since practitioners often deploy small to quite powerful on-prem hardware (e.g., SBCs, H100-class GPUs as such) and still describe these as edge setups. Clarifying the target hardware and resource envelope would make the paper impact to certain group with more focused approach.
>
> More importantly, I do not find the claim of being the *first* systematic investigation of single-stream lifelong generative learning fully convincing. There is a substantial line of work on **GANs and VAEs with replay** for online / continual learning, and the recent paper *Learning from One Continuous Video Stream* by Carreira et al. explicitly studies autoregressive generative models trained on a single, continuous stream. In that light, this submission is best viewed as the **first diffusion-based instance** of this general idea, not the first exploration of the regime itself. The core mechanism—experience replay to mitigate forgetting in a streaming setting—is well established.
>
> Given that positioning, a foundational paper in this space should, in my view, include **in-depth comparisons to existing generative approaches** (GANs, VAEs, or at least the most closely related autoregressive work), even if all methods are trained at similar small/medium scales. Currently the paper restricts comparisons to variants of diffusion trained from scratch, which keeps the study rather self-contained.
>
> ---
>
> ### 2. Datasets and benchmarks (Ego4D-Stream)
>
> The authors note that their datasets include driving and Minecraft domains and that the 1M-frame single-stream construction is designed to stress long, autocorrelated training, which typical clip-based benchmarks do not support. I agree that the custom datasets are useful, and I did look at the provided video snippets.
>
> However, for a study that aspires to be foundational, I still believe it is important to **anchor the method on at least one standard benchmark**. Otherwise it is difficult to judge how the observed behavior relates to the broader literature on video generative models and continual learning. A natural candidate, and one that is already cited in the paper, is **Ego4D-Stream**. Even a limited experiment—e.g., training the proposed lifelong diffusion setup on a subset of Ego4D-Stream sequences—would help connect the results to a widely recognized egocentric streaming dataset and allow more direct comparison to prior work.
>
> ---
>
> ### 3. Analysis and interpretation of results
>
> The authors argue that their experiments across several streams, architectures, parameter scales, and optimizers constitute a broad empirical study, and that the main insight is that “video diffusion models can be trained online from a single stream while achieving performance comparable to offline training.” I still find this conclusion too strong, for several reasons:
>
> - The **metric differences between Offline and Lifelong** are typically small but systematic, and on more complex streams (Drive, PLAICraft) Offline often performs noticeably better. The rebuttal reiterates that neither regime consistently dominates, but does not provide deeper analysis of *why* the Lifelong regime underperforms in some settings, or under what conditions one would expect it to match or surpass Offline training.
> - The explanation that replay helps by re-exposing the model to past data, whereas Orthogonal AdamW only addresses gradient autocorrelation, is qualitatively reasonable but remains quite high-level. It does not rise to the level of a detailed analytical or empirical study of the training dynamics.
> - The evaluation relies entirely on FVD/JEDi/minADE/ColorKL. These are standard and useful metrics, but they are known to be imperfect proxies for perceptual quality. Some of the sample videos exhibit visible noise or distortion even when metric differences are modest. A **human-in-the-loop evaluation** or additional perceptual metrics would strengthen the claim that Lifelong performance is truly “comparable” to Offline in practice.

---

> ### Author Response · Authors · 2025-12-05
>
> We thank the reviewer for acknowledging the value of our datasets and for understanding the relevance of from-scratch training of video diffusion models on single video streams for embodied agent-like settings. We address the remaining concerns in detail below.
>
> > I do not find the claim of being the first systematic investigation of single-stream lifelong generative learning fully convincing. There is a substantial line of work on GANs and VAEs with replay for online / continual learning, and Learning from One Continuous Video Stream [1] studies autoregressive generative models trained on a single, continuous stream. In this light, the submission is best viewed as the first diffusion-based instance of this general idea.
>
> We believe our claim is justified for the following reasons:
>
> 1. Distinction from Learning from One Continuous Video Stream [1]: Their evaluated model is a discriminative sequence model that returns a single prediction of the future frames. In contrast, our video diffusion models generate a diverse and realistic set of future frames, which we extensively demonstrate in our qualitative results. The ability to sample possible futures instead of returning a single prediction is important for world-modeling applications [2,3]. Furthermore, their training streams are formed by concatenating short-to-medium independent video clips, which introduces abrupt context shifts resembling i.i.d. training to their video streams. Our proposed datasets consist of true continuous video streams, providing the long-horizon continuity that embodied agents experience.
>
> 2. GANs and VAEs in continual learning: To our knowledge, there is no prior work on streaming (online) training of GANs or VAEs in the video domain. The closest works [4,5] address offline continual learning, where video VAE models are updated on a sequence of datasets and can be trained on until convergence.
>
> Given the differences in model class (generative vs. discriminative), learning setting (streaming vs. offline continual learning), and data regime (continuous sensor stream vs. clip concatenation), we believe our characterization as the first systematic investigation of single-stream lifelong generative learning is accurate.
>
> > Compare with existing generative methods on the same data: Include at least one baseline from prior streaming/continual generative work (e.g., the most relevant autoregressive model from Learning from One Continuous Video Stream) trained on your single-stream datasets.
>
> We note that we have already baselined Orthogonal AdamW [6] for video diffusion model training, which is a follow-up work that improves on the results from Carreira et al [1].
>
> > Evaluate on a standard streaming benchmark: For example, apply your lifelong diffusion setup to a subset of Ego4D-Stream, which has already been used to study autoregressive generative models in a continuous video regime. It is not necessary to run every baseline there, but seeing how your approach behaves on this dataset would make the results much more transferable.
>
> We appreciate the suggestion to evaluate on existing streaming benchmark. However, we believe such an evaluation is not critical for our work for the following reasons.
>
> 1. Mismatch with our problem setting: Our focus is on single, continuous video streams without discontinuities. “Streaming” benchmarks, including those constructed from Ego4D, are formed by concatenating many short/medium clips, which introduces abrupt semantic jumps.
>
> 2. Accessibility constraints and sparsity of streaming benchmark with extensive diffusion evaluation: The referenced dataset Ego4D-Stream [1] is not publicly available, even though a larger database of Ego4D is. This limits its suitability as a reproducible benchmark. Moreover, we are not aware of any widely used streaming video benchmarks for training video diffusion models in particular.
>
> 3. Breadth and relevance of our evaluation: Our paper already evaluates in-stream and out-of-stream behavior across five heterogeneous continuous-video datasets (Tables 1–5), covering real-world perceptual data and complex multi-agent dynamics. These datasets directly align with the specific lifelong generative learning scenario we study, and we have found them sufficient to analyze the phenomena central to our work.
>
> For these reasons, we believe the current evaluation is well aligned with the goals of the paper, and that adopting non-continuous clip-concatenated benchmarks would provide limited additional insight for the setting we address.

---

> ### Author Response · Authors · 2025-12-05
>
> > Add human-centric or stronger perceptual evaluation: Consider a small human study or additional perceptual metrics to better assess qualitative differences between Offline and Lifelong training.
>
> To address this suggestion, we conducted a **human evaluation for the rebuttal** comparing future frames generated by lifelong-trained versus offline-trained models. We extracted 76 clip pairs from the supplementary material (4 generations × 19 scenarios) that illustrate key characteristics of the datasets. For every pair, five independent Amazon Mechanical Turk participants were asked, without knowing which model produced which clip, to indicate whether the left or right clip was of higher quality, or whether they appeared similar.
>
> The aggregated preferences across datasets are shown below:
>
> | **Dataset**                 | **Offline Better** | **Lifelong Better** | **Similar** |
> | --------------------------- | ------------------ | ------------------- | ----------- |
> | Lifelong Bouncing Balls (O) | 24                 | 25                  | 11          |
> | Lifelong Bouncing Balls (C) | 23                 | 26                  | 11          |
> | Lifelong 3D Maze            | 19                 | 25                  | 16          |
> | Lifelong Drive              | 43                 | 41                  | 16          |
> | Lifelong PLAICraft          | 43                 | 40                  | 17          |
>
> Across all datasets, participants show no consistent preference for either offline or lifelong training. Even when excluding responses marked “similar,” votes remain balanced between the two methods, indicating that the perceptual quality of lifelong-trained diffusion models is on par with that of their offline counterparts.
>
>
> > The metric differences between Offline and Lifelong are typically small but systematic, and on more complex streams (Drive, PLAICraft) Offline often performs noticeably better.
>
> We respectfully disagree that Offline Learning performs noticeably better over Lifelong Learning on more complex streams like Drive and PLAICraft. For example, on Lifelong PLAICraft results in Table 5, Offline Learning generally performs very slightly better for U-Net models, but the trend is reversed for the Transformer models. Even on Lifelong Drive results in Table 4, Offline Learning and Lifelong Learning performance differences are very close, and their error bars overlap in some cases. Importantly, any numerical gap between the two configurations is overshadowed by the effect of changing model architecture and size. Lastly, our qualitative evaluations further reinforce our claim that the performance differences are comparable. Overall, the evidence does not support a clear advantage for Offline Learning on these complex datasets.
>
>
>
> > Provide more in-depth analysis of why the method behaves as it does: For example, analyze how replay buffer size, non-stationarity level, and model capacity interact.
>
> Our paper identifies catastrophic forgetting as the primary challenge in our lifelong learning setting: buffer-free methods perform poorly even on simple non-stationary data such as Lifelong Bouncing Balls (C), and streaming AdamW exhibits progressively degraded performance on earlier frames (Fig. 4). Orthogonal AdamW also underperforms in this regime, reinforcing that forgetting, not gradient autocorrelation, is the dominant issue.
>
> We then establish that experience replay substantially alleviates this issue. Specifically, replaying 5% of the stream is sufficient for Balls and Mazes, while 20% is sufficient for the more complex Drive and PLAICraft streams, to bring their performance to a comparable level to offline training and 100% memory experience replay. To extend this analysis, **we provide additional rebuttal experiments** examining smaller replay buffers (5%) for Drive and PLAICraft using U-Net diffusion models:
>
> | Dataset | Method | Train FVD | Train JEDi | Train Loss | Test FVD | Test JEDi | Test Loss |
> |-|-|-|-|-|-|-|-|
> | Lifelong Drive | 0% Replay | 322.8 | 0.608 | 0.041 | 183.4 | 0.343 | 0.029 |
> | Lifelong Drive | 5% Replay | 29.3 | 0.131 | 0.039 | 33.2 | 0.166 | 0.028 |
> | Lifelong Drive | 20% Replay | 17.0 | 0.072 | 0.030 | 21.8 | 0.106 | 0.027 |
> | Lifelong PLAICraft | 0% Replay | 239.5 | 2.038 | 0.047 | 270.6 | 2.280 | 0.050 |
> | Lifelong PLAICraft | 5% Replay | 69.8 | 0.842 | 0.036 | 142.4 | 1.200 | 0.045 |
> | Lifelong PLAICraft | 20% Replay | 62.9 | 0.876 | 0.034 | 130.8 | 1.189 | 0.042 |
>
> We find that while there is a general performance dropoff compared to the 20% memory experience replay, the dropoff is minor compared to that of the 0% memory experience replay setup. This suggests that, owing to dataset complexity and non-stationarity, the replay buffer size where additional capacity does not significantly improve the model performance is larger for Lifelong Drive and Lifelong PLAICraft than for Lifelong Bouncing Balls and Lifelong 3D Maze.

---

> > ### Author Response · Authors · 2025-12-05
> >
> > > It would help, however, if the paper gave a more concrete definition of the edge systems it targets. Clarifying the target hardware and resource envelope would make the paper impact to certain group with more focused approach.
> >
> > Thank you for the suggestion. Because edge-device hardware varies widely, the compute–memory tradeoff is application dependent. We agree that clarifying the intended resource envelope would improve clarity, and we will add this to the camera-ready draft.
> >
> > We appreciate the reviewer’s thoughtful engagement with our rebuttal and hope that our responses address all outstanding concerns. We would be happy to clarify any remaining questions.
> >
> > ### References
> >
> > [1] Carreira, J., King, M., Patraucean, V., Gokay, D., Ionescu, C., Yang, Y., ... & Zisserman, A. (2024). Learning from one continuous video stream. In Proceedings of the IEEE/CVF Conference on Computer Vision and Pattern Recognition (pp. 28751-28761).
> >
> > [2] Alonso, E., Jelley, A., Micheli, V., Kanervisto, A., Storkey, A. J., Pearce, T., & Fleuret, F. (2024). Diffusion for world modeling: Visual details matter in atari. Advances in Neural Information Processing Systems, 37, 58757-58791.
> >
> > [3] Hafner, D., Pasukonis, J., Ba, J. et al. Mastering diverse control tasks through world models. Nature 640, 647–653 (2025). https://doi.org/10.1038/s41586-025-08744-2
> >
> > [4] Campo, D., Slavic, G., Baydoun, M., Marcenaro, L., & Regazzoni, C. (2020, October). Continual learning of predictive models in video sequences via variational autoencoders. In 2020 IEEE International Conference on Image Processing (ICIP) (pp. 753-757). IEEE.
> >
> > [5] Chen, G., Zhang, W., Lu, H., Gao, S., Wang, Y., Long, M., & Yang, X. (2022). Continual predictive learning from videos. In Proceedings of the IEEE/CVF Conference on Computer Vision and Pattern Recognition (pp. 10728-10737).
> >
> > [6] Han, T., Gokay, D., Heyward, J., Zhang, C., Zoran, D., Patraucean, V., Carreira, J., Damen, D., & Zisserman, A. (2025). Learning from Streaming Video with Orthogonal Gradients. In Proceedings of the IEEE/CVF Conference on Computer Vision and Pattern Recognition (CVPR) (pp. 13651-13660).

---

### Comment · Action_Editor_nF1o · 2025-11-29
**Discussions between authors and reviewers**

Dear authors,

Please read through the reviews and any requested changes. The discussion period lasts two weeks, and a few days have already passed. I kindly ask that you start your responses as soon as possible to help facilitate a constructive discussion.

---

### Author Response · Authors · 2025-12-14
**Rebuttal Summary**

We thank all reviewers for their thoughtful feedback. Across reviews, the main discussion points were: (i) clarifying our key claim and positioning relative to prior work and (ii) strengthening the experimental validation and ablations.

In response, we refined our positioning and made targeted updates that directly address these concerns. First, we sharpened the paper’s scope and novelty: our main contribution is a proof-of-concept demonstrating that video diffusion models can be lifelong learned from scratch on an in-order single-video stream while achieving performance comparable to shuffled i.i.d. training. To support this claim, we introduced four new datasets, which are the first in the lifelong learning community to contain 1 million frames each from a single continuous video stream without abrupt context switches, and evaluated multiple lifelong learning algorithms on them. These experiments show that simple experience replay, retaining 5–20% of the video stream frames, is sufficient to maintain competitive performance with i.i.d. training, whereas buffer-free approaches can be prone to catastrophic forgetting even on a toy non-stationary data stream. We also explicitly contrast our work with prior approaches on discriminative video model training on concatenated video streams.

Second, we strengthened the empirical evidence and ablations. Following suggestions from reviewers QeqN and jxYM, the rebuttal reports results with 5% replay buffer size for Lifelong Drive and Lifelong PLAICraft, clarifying the threshold at which replay matches i.i.d. performance. We further address reviewer jxYM’s concerns regarding long-horizon video generation by demonstrating that our core finding holds even when generating videos autoregressively at lengths four times greater than the model’s context window in the new rebuttal results. Finally, responding to reviewer QeqN’s, we conducted a human evaluation study showing that videos produced by offline-trained and lifelong-trained diffusion models are indistinguishable to humans across all datasets.

Overall, the added experiments and clarifications directly resolve the reviewers’ core concerns and reinforce the main takeaway: training autoregressive video diffusion models from scratch on a single video stream—resembling the experience of embodied agents—is not only possible, but can also be as effective as standard offline training given the same number of gradient steps across a diverset set of video streams, model classes, and optimizers. Since video generative modeling is a key component of vision-based world models, we expect our findings and datasets to be of interest to future work on continually updating embodied systems.

---

### Decision · Action_Editor_nF1o · 2025-12-28

**Recommendation:** Reject

**Additional Comments:**

The reviewers (specifically QeqN and jxYM) raised fundamental concerns that could not be fully resolved during the rebuttal phase. To warrant acceptance in TMLR, a future revision would need to:
- Strengthen the Motivation: Shift the focus from storage efficiency to more critical bottlenecks like compute efficiency or rapid adaptation.
- Expand Evaluation: Validate the method on standard community benchmarks to prove transferability and allow direct comparison with prior art.
- Demonstrate stricter constraints: Show that the method remains effective with significantly smaller replay buffers (e.g., <1%) to justify the "lifelong learning" claim, or provide a deeper theoretical analysis of the trade-offs involved.
- Broaden Baselines: Include comparisons against a wider range of continual learning strategies and architectures.

**Audience:**

Yes

**Audience Explanation:**

the core problem addressed: training video diffusion models on continuous, non-i.i.d. data streams, is highly relevant to the TMLR audience, particularly those working at the intersection of generative AI and embodied systems (robotics). The proposed datasets (Lifelong Bouncing Balls, 3D Maze, Drive, and PLAICraft) represent a potentially valuable resource for researchers studying long-horizon temporal dynamics and online world modeling. Furthermore, the empirical investigation into the stability of diffusion models trained from scratch on correlated streams offers a useful reference point for future work in this emerging area.

**Claims And Evidence:**

No

**Claims Explanation:**

While the authors present extensive experimental data, the evidence does not convincingly support the broad claims regarding the efficacy and relevance of the proposed "lifelong learning" approach.
- Definition of Lifelong Learning: As noted by Reviewer jxYM, the method requires retaining a very large percentage of the video stream (up to 20%) in the replay buffer to match offline performance. This significant data retention blurs the distinction between lifelong learning and offline training, failing to convincingly demonstrate a solution that works under the strict memory constraints typically associated with lifelong or embodied learning.
- Benchmarking Limitations: Reviewer QeqN highlighted that the evaluation relies exclusively on custom datasets introduced by the authors. The lack of evaluation on standard streaming benchmarks (e.g., Ego4D or standard video generation benchmarks) or comparison against broader baselines (such as pre-trained models) limits the ability to verify the generalizability of the claims.
- Scope of Comparisons: The comparisons are restricted to the authors' own baselines (Experience Replay vs. No Replay). The absence of comparisons to other generative continual learning strategies (e.g., GANs/VAEs with replay) or rigorous analysis of why simple replay suffices limits the strength of the evidence.

**Resubmission Of Major Revision:**

The authors may consider submitting a major revision at a later time.